# Segment-driven Structural Induction and Semantic Alignment for Heterogeneous Tabular Representation

**Woojun Jung** [1]  **Susik Yoon** [1]

## Abstract

Real-world domains often contain heterogeneous tables whose headers vary while their underlying attribute semantics are shared, making it difficult to induce domain-specialized semantics from table-local evidence alone. Existing encoders model parts of this problem, but often underuse column-level value distributions and apply uniform objectives across attributes with different semantic roles. We propose NAVI, a segment-centric pretraining framework that treats each header–value pair as the unit for aggregating schema-level structural evidence and column-level distributional evidence. We realize this design through Masked Segment Modeling and Entropy-driven Segment Alignment, which jointly enforce structured header–value coupling and semantic alignment across stable and instance-specific attributes. Experiments on heterogeneous in-domain tables show improved reconstruction, semantic consistency, and downstream utility across evaluation settings overall. The source code of NAVI is available at: https://github.com/woojoonjung/NAVI/.

## 1. Introduction

Tables are among the most prevalent modalities in a modern society, serving as the primary interface for structured information in diverse domains, ranging from e-commerce to media platforms. Real-world tabular data in these domains is rarely confined to a single table; rather, it usually spans a collection of heterogeneous tables that collectively define shared domain knowledge. While these in-domain tables differ in their schema, they are frequently analyzed jointly to support downstream, domain-specific applications.

[1]Department of Computer Science & Engineering, Korea University, Seoul, South Korea. Correspondence to: Susik Yoon <susik@korea.ac.kr>.

*Proceedings of the 43rd International Conference on Machine Learning*, Seoul, South Korea. PMLR 306, 2026. Copyright 2026 by the author(s).

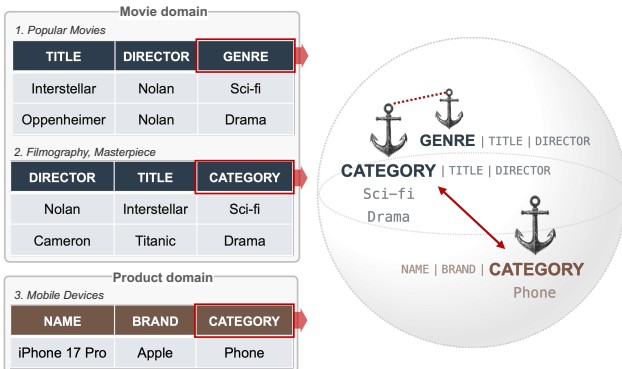

*Figure 1.* Semantically similar attributes may appear under different headers, while identical headers may correspond to different attribute semantics across domains.

Unlike unstructured text, where semantics are primarily conveyed through token composition and linguistic context, tables organize semantics around *attributes* that are typically realized through table headers. Within a domain, attribute semantics should be understood as domain-specialized semantic posteriors inferred from evidence accumulated across heterogeneous tables. The primary challenge in this setting is that these semantics cannot be treated as fixed lexical meanings inferred directly from header names alone.

Instead, such semantics emerge from two complementary sources of contextual evidence. First, *schema context* provides structural evidence through co-occurring headers. For example, headers such as TITLE, DIRECTOR, and GENRE collectively indicate a movie-related schema. Second, *column context* provides distributional evidence through empirical value realizations accumulated across rows and tables. In Figure 1, the interpretation of CATEGORY depends on whether its values correspond to movie genres (e.g., Drama) or product types (e.g., Phone).

Moreover, attributes can assume heterogeneous semantic roles across tables. Some attributes correspond to stable domain-coherent properties, while others primarily distinguish individual entities. For example, DIRECTOR acts as a shared domain property in 'Popular Movies', but behaves more like an entity-identifying attribute in 'Filmography'. This suggests that column-level evidence should not contribute uniformly when refining attribute semantics.

Existing tabular learning methods only partially address these challenges. Structure-aware encoders (Herzig et al., 2020; Yin et al., 2020; Iida et al., 2021; Deng et al., 2022) primarily focus on contextual interactions within individual tables, but often assume that header semantics are locally recoverable from table structure alone. More recent approaches (Ye et al., 2024; Yan et al., 2024; Jung & Yoon, 2025) improve robustness to heterogeneous feature types or schema variation, yet still treat headers largely as surface-level schema tokens without explicitly modeling how attribute semantics emerge jointly from schema-level and column-level evidence accumulated across tables. Furthermore, existing methods apply uniform representation objectives across columns, overlooking the heterogeneous semantic roles that attributes may assume across tables.

Motivated by these limitations, we propose NAVI (ENtropy-driven Alignment with Header–Value Induction), a dual-context pretraining framework for heterogeneous tabular representation learning. The core abstraction of NAVI is the *segment*, a symbolic header–value realization that serves as the fundamental unit for aggregating structural and distributional evidence across tables. NAVI instantiates this idea through (i) *Masked Segment Modeling*, which captures schema-level structural context from co-occurring header–value realizations, and (ii) *Entropy-driven Segment Alignment*, which models column-level distributional context through role-aware contrastive alignment.

To our knowledge, NAVI is the first segment-centric framework to jointly leverage schema-level structural evidence and column-level distributional evidence, enabling attribute semantics to be refined into domain-specialized representations across heterogeneous in-domain tables. Furthermore, NAVI introduces a conceptual framework that explicitly distinguishes latent attribute semantics, domain attributes, and table-specific header realizations, providing a principled basis for interpreting semantic consistency across heterogeneous schemas. Extensive experiments demonstrate that NAVI consistently improves semantic consistency, robustness, and downstream representation quality over existing table representation methods.

## 2. Related Work

Existing works have studied tabular learning from diverse perspectives, ranging from tree-based learners (Chen & Guestrin, 2016), pretrained neural architectures for prediction (Hollmann et al., 2023), and language model (LM)-based feature engineering (Lim & Yoon, 2025). Recently, the symbolic and lexical information prevalent in many real-world tables has accelerated the adaptation of Transformer-based LMs to tabular data (Fang et al., 2024; Badaro et al., 2023), leveraging contextual semantic priors learned from unstructured text (Vaswani et al., 2017; Devlin et al., 2019).

### 2.1. Structure-aware Table Encoders

Early attempts to represent tables with LMs addressed the structural discrepancy between natural language and tables by adapting Transformer architectures like BERT to navigate spatial hierarchies. Initial efforts such as TAPAS and TaBERT (Herzig et al., 2020; Yin et al., 2020) inject table structure into LMs through role embeddings or vertical attention, enabling joint contextualization of natural language and tabular inputs. Subsequent works such as TABBIE, TUTA, and TURL (Iida et al., 2021; Wang et al., 2021; Deng et al., 2022) further emphasize intrinsic table structure, modeling row–column dependencies or relational constraints. However, these methods infer header semantics strictly within individual tables. Consequently, they fail to distinguish latent domain attributes from table-specific header realizations, limiting their ability to refine stable attribute semantics across heterogeneous in-domain tables.

### 2.2. Feature-oriented Table Encoders

A prominent paradigm focuses on modeling tabular semantics at a unified feature level. On one hand, TP-BERTa (Yan et al., 2024) addresses data type heterogeneity, employing magnitude tokenization and intra-feature attention to map continuous numerical values alongside textual categorical data into a shared discrete embedding space. Conversely, CM2 (Ye et al., 2024) prioritizes cross-table scalability, mapping headers and cells directly into unified text phrases for permutation-invariant masked table modeling. Despite such improvements, these methods still do not explicitly distinguish domain-level attribute semantics from their table-specific header semantics, limiting their ability to refine domain-specialized semantics across heterogeneous tables. Moreover, they apply uniform representation objectives across columns, overlooking the heterogeneous semantic roles that attributes may assume across tables.

### 2.3. Domain-aware Table Encoders

Recent work moves toward domain-aware representations, emphasizing transferable attribute semantics across in-domain tables rather than treating headers as table-confined semantic units. Specifically, HAETAE (Jung & Yoon, 2025) introduces header-anchored pretraining, treating schemas as shared global structures within a domain. It models tabular semantics primarily through schema context and separately encodes headers to preserve stable schema priors. While this improves semantic stability, it does not explicitly accumulate column-level evidence across tables to refine domain-specialized attribute semantics, nor does it account for the heterogeneous semantic roles attributes assume. These limitations motivate a framework that jointly models schema and column context while selectively aligning column signals according to their semantic role.

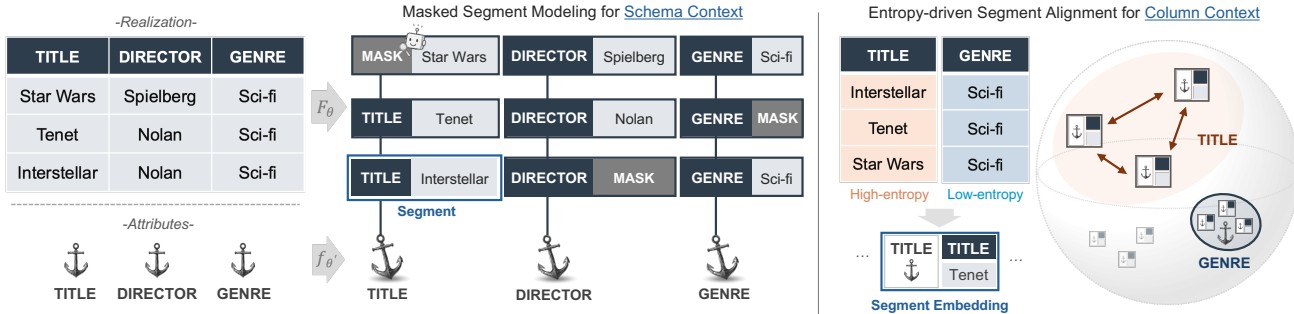

*Figure 2.* Overall procedure of NAVI. We jointly optimize NAVI with masked segment modeling and entropy-driven segment alignment.

## 3. Methodology

As shown in Figure 2, NAVI models heterogeneous tabular semantics through two complementary segment-centric learning processes. The upper branch captures *schema context* via masked segment modeling over schema-preserving row serialization, enabling structural dependencies among co-occurring header–value realizations to be learned. The lower branch captures *column context* through entropy-driven segment alignment, where segment embeddings are contrastively aligned according to the empirical distributional characteristics of their associated attributes. Together, these components refine attribute semantics through structural schema evidence and distributional column evidence accumulated across heterogeneous tables.

### 3.1. Dual-context Representation Framework

We distinguish between domain attributes, their contextual realizations, and the latent semantic concepts they represent in heterogeneous tables. Let $\mathcal{A}$ denote the set of domain attributes, where each attribute $a \in \mathcal{A}$ is represented by a canonical textual identifier shared across tables. Headers are viewed as table-specific schema realizations of domain attributes. Our objective is to recover latent domain concepts from contextual evidence accumulated across tables.

**Schema and Column Contexts.** We model attributes through two complementary contexts. *Schema context* refers to contextual evidence involving schema elements (i.e., headers). In this work, we operationalize schema context through row-wise header–value co-occurrence patterns. For each table $t \in \mathcal{T}$, let $\mathcal{H}_t = \{h_1, \ldots, h_K\}$ denote its observed schema. Each header $h_k \in \mathcal{H}_t$ is associated with an attribute $a_k \in \mathcal{A}$. A row $r \in t$ represents an entity as:

$$r = \{(h_k, v_{r,k})\}_{k=1}^{K},$$

where $v_{r,k}$ denotes the value associated with header $h_k$ in row $r$. By jointly modeling headers and their corresponding value realizations within rows, schema context provides structural evidence for refining attribute semantics.

*Column context* captures semantic regularities induced from empirical value distributions accumulated across rows and tables. For a domain attribute $a_k \in \mathcal{A}$ observed in table $t$, we define the corresponding column context as:

$$\mathcal{C}_t(a_k) = \{v_{r,k} \mid r \in t\}.$$

Together, schema and column contexts provide complementary structural and distributional evidence for understanding attribute semantics in heterogeneous tables.

**Segment Formulation.** To operationalize schema and column contexts, we define a *segment* as the fundamental symbolic unit corresponding to a header–value pair within a table. Segments serve as the basic units through which structural and distributional evidence are aggregated across rows and tables. Unlike isolated cells, segments preserve the functional dependency between headers and values, enabling them to be modeled jointly.

We denote by $\mathbf{x}(\cdot)$ a tokenization function that maps an input text into a sequence of tokens. In particular, $\mathbf{x}(r)$ denotes the serialized token sequence constructed from row $r$.

**Definition 3.1** (Segment). For a domain attribute $a_k \in \mathcal{A}$ and its associated header–value realization $(h_k, v_{r,k})$ in row $r$, we define a segment as:

$$\mathbf{s}_{r,k} = [\mathbf{x}(h_k), \, \colon, \mathbf{x}(v_{r,k})].$$

Segment $\mathbf{s}_{r,k}$ represents the symbolic realization of a domain attribute $a_k$ within a table.

**Attribute and Realization Embeddings.** To model segments under schema and column contexts, we encode domain attributes and their table-specific realizations in a shared embedding space using two separate encoders. A lightweight encoder $f_{\theta'}$ independently maps attributes to context-free embeddings, while a Transformer encoder $F_\theta$ contextualizes header–value realizations within serialized rows. We next define the corresponding attribute and realization embeddings. Implementation details of $f_{\theta'}$ and $F_\theta$ are provided in Appendix D.1.

**Definition 3.2** (Context-free Attribute Embedding). For each domain attribute $a \in \mathcal{A}$, we define its context-free embedding using a lightweight encoder $f_{\theta'}$ as:

$$\mathbf{e}_a = \text{Pool}\big(f_{\theta'}(\mathbf{x}(a))\big),$$

where $\text{Pool}(\cdot)$ denotes mean pooling and $f_{\theta'}$ is applied independently of any table instance.

**Definition 3.3** (Contextualized Realization Embedding). Given a tokenized row $\mathbf{x}(r)$ with length $L$, a Transformer encoder $F_\theta$ produces contextualized token embeddings:

$$[e_0, \ldots, e_{L-1}] = F_\theta(\mathbf{x}(r)). \tag{1}$$

Let $\mathcal{S}_{r,k}^{(h)}$ and $\mathcal{S}_{r,k}^{(v)}$ denote the token spans corresponding to header $h_k$ and value $v_k$, respectively. We define the contextualized header and value representations as:

$$\mathbf{e}_{r,k}^{(h)} = \text{Pool}(\{e_t \mid t \in \mathcal{S}_{r,k}^{(h)}\}),$$
$$\mathbf{e}_{r,k}^{(v)} = \text{Pool}(\{e_t \mid t \in \mathcal{S}_{r,k}^{(v)}\}).$$

### 3.2. Masked Segment Modeling

**Schema-preserving Serialization.** Given a row $r = \{(h_k, v_k)\}_{k=1}^K$, we construct its token sequence $\mathbf{x}(r)$ by concatenating segments into a linearized format:

$$\mathbf{x}(r) = \big[[\texttt{CLS}], \mathbf{s}_{r,1}, [\texttt{SEP}], \ldots, \mathbf{s}_{r,K}, [\texttt{SEP}]\big],$$

where each segment is delimited by a separator token (e.g., [SEP]). By serializing rows as sequences of segments, the encoder can model schema context through interactions among co-occurring header–value realizations. The serialization process is exemplified as follows:

**Schema-aware Contextualization.** To promote schema-level stability and reduce sensitivity to column permutations, we reinitialize positional embeddings within segment boundaries and augment token embeddings with the corresponding attribute embedding. Let $e_{\text{word}}(\cdot)$ denote the token embedding function and $P_j$ denote learnable positional embeddings. For a token $x_t$ belonging to segment $\mathbf{s}_{r,k}$ associated with attribute $a_k$, we define:

$$e_{\text{pos}}(x_t) = P_j, \quad j = 0, \ldots, l_k,$$
$$e_t = e_{\text{word}}(x_t) + e_{\text{pos}}(x_t) + \mathbf{e}_{a_k},$$

where $l_k$ is the number of tokens in segment $\mathbf{s}_{r,k}$. Here, $x_t$ denotes the $t$-th token in $\mathbf{x}(r)$, and $a_k \in \mathcal{A}$ is the attribute corresponding to header $h_k$, with $\mathbf{e}_{a_k}$ its attribute

embedding. The encoder $F_\theta$ processes these embeddings to produce token representations as in Eq. (1). This formulation injects attribute priors into all tokens within a segment, enabling consistent schema-aware contextualization.

**Structure-aware Masking.** Standard masked language modeling (MLM) has proven effective for natural language (Devlin et al., 2019), but its direct application to tables is suboptimal because it treats schema and value tokens uniformly. Headers represent attribute realizations, while values instantiate these attributes with instance-specific information. To explicitly model these dependencies, we introduce a masked segment modeling (MSM) that explicitly models schema–value dependencies by partitioning each row of segments into three masking regimes:

- **Header-masked segments:** header tokens in selected segments are masked, forming the set $\mathcal{M}_h$. The model must recover header names from associated values.

- **Value-masked segments:** value tokens in selected segments are masked, forming the set $\mathcal{M}_v$. The model must infer values from headers and row context.

- **Vanilla MLM:** a random subset of remaining tokens is masked, forming $\mathcal{M}_r$. This regularizes the model to prevent overfitting header–value co-occurrences.

**Objective.** For each masked token $m \in \mathcal{M}$ with contextualized token embedding $e_t$, the classifier with parameters $\{W, b\}$ produces a logit vector $\mathbf{z}_m = W e_m + b \in \mathbb{R}^{|\mathcal{V}|}$. The masked segment modeling (MSM) loss is then given by the standard softmax cross-entropy:

$$\mathcal{L}_{\text{msm}} = -\frac{1}{|\mathcal{M}|} \sum_{m \in \mathcal{M}} \log \frac{\exp(\mathbf{z}_m[m])}{\sum_{v \in \mathcal{V}} \exp(\mathbf{z}_m[v])},$$

where $\mathbf{z}_m[v]$ is the logit corresponding to item $v$ of vocabulary $\mathcal{V}$ at the position of $m$, and $\mathcal{M}$ is the union of $\mathcal{M}_h, \mathcal{M}_v, \mathcal{M}_r$. The MSM objective, coupled with structured masking, compels the encoder to capture the functional roles of tokens and header-value dependencies within the tabular schema.

### 3.3. Entropy-driven Segment Alignment

Contrastive learning (Oord et al., 2018; Chen et al., 2020; Lee et al., 2022) is widely used to organize representations according to a target semantic objective. Building on symbolic segments, we construct *segment embeddings* that integrate context-free attribute semantics with contextualized header and value realizations. By aligning these embeddings in a distribution-aware manner, we enable attribute semantics to adapt to table-specific statistics while preserving semantic consistency across heterogeneous tables.

**Definition 3.4** (Segment Embeddings). For a row $r \in t$ and a segment $\mathbf{s}_{r,k}$ associated with domain attribute $a_k \in \mathcal{A}$, we define the corresponding segment embedding as:

$$\sigma_{r,k} = g\big(\mathbf{e}_{a_k} \,\|\, \mathbf{e}_{r,k}^{(h)} \,\|\, \mathbf{e}_{r,k}^{(v)}\big),$$

where $\mathbf{e}_{a_k}$ is the context-free attribute embedding in Definition 3.2, $\mathbf{e}_{r,k}^{(h)}$ and $\mathbf{e}_{r,k}^{(v)}$ are the contextualized header and value embeddings in Definition 3.3, $\|$ denotes concatenation, and $g(\cdot)$ is a trainable two-layer feedforward network with GELU activation and layer normalization.

**Entropy-based Attribute Categorization.** To capture the heterogeneous roles of attributes within a table, we categorize them based on the empirical distribution of their column values. For each table $t \in \mathcal{T}$ and attribute $a \in \mathcal{A}^t$ observed in $t$, we compute the normalized entropy of its column:

$$\mathrm{H}_t(a) = - \sum_{v \in \mathcal{C}_t(a)} p_t(v \mid a) \log p_t(v \mid a),$$

where $p_t(v \mid a)$ denotes the empirical frequency of value $v$ in its column $\mathcal{C}_t(a)$. Using fixed quantile thresholds $\theta_{\text{low}}$ and $\theta_{\text{high}}$, we partition $\mathcal{A}^t$, the set of domain attributes observed in table $t$, into two disjoint subsets:

$$\mathcal{A}_{\text{low}}^t = \{a \in \mathcal{A}_t \mid \mathrm{H}_t(a) \leq \theta_{\text{low}}\},$$
$$\mathcal{A}_{\text{high}}^t = \{a \in \mathcal{A}_t \mid \mathrm{H}_t(a) \geq \theta_{\text{high}}\}.$$

Low-entropy attributes ($\mathcal{A}_{\text{low}}^t$) correspond to stable, domain-coherent properties (e.g., *genre*), while high-entropy attributes ($\mathcal{A}_{\text{high}}^t$) capture instance-specific variations (e.g., *title*). This categorization is computed independently for each table and remains fixed during training.

**Batch Construction.** Training proceeds over stratified batches of rows sampled from multiple tables within the same domain. At each step, we sample a set of tables $\mathcal{T}_{\mathcal{B}} \subset \mathcal{T}$ and construct a batch $\mathcal{B}$ by drawing approximately $|\mathcal{B}|/|\mathcal{T}_{\mathcal{B}}|$ rows from each table.

Each row $r \in \mathcal{B}$ is associated with its source table $\mathrm{t}(r)$, and we denote $\mathcal{B}^t = \{r \in \mathcal{B} \mid \mathrm{t}(r) = t\}$ as the subset of rows originating from table $t$. During training, segments from each row is aligned according to the attribute partition $(\mathcal{A}_{\text{low}}^t, \mathcal{A}_{\text{high}}^t)$ of its source table.

**Learning Objective.** Given a query $q$, a positive sample $x^+$, a set of negative samples $\mathcal{X}^-$, and a temperature $\tau$, the InfoNCE objective (Oord et al., 2018) is defined as:

$$\mathcal{L}_{\text{NCE}}(q, x^+, \mathcal{X}^-, \tau)$$
$$= -\log \frac{e^{\text{sim}(q,x^+)/\tau}}{e^{\text{sim}(q,x^+)/\tau} + \sum_{x^- \in \mathcal{X}^-} e^{\text{sim}(q,x^-)/\tau}}.$$

For attributes $a_k \in \mathcal{A}_{\text{low}}^t$, we align segment embeddings with attribute embeddings, encouraging semantically consistent attribute representations across rows and tables:

$$\mathcal{L}_{\text{low}}^t = \mathbb{E}_{r \sim \mathcal{B}^t, \, a_k \in \mathcal{A}_{\text{low}}^t}\Big[\mathcal{L}_{\text{NCE}}\big(\sigma_{r,k}, \, \mathbf{e}_{a_k}, \, \mathcal{E}_{\text{low}}^-, \, \tau_{\text{low}}\big)\Big],$$
$$\mathcal{E}_{\text{low}}^- = \{\, \mathbf{e}_{a'} \mid a' \in \mathcal{A}_{\text{low}}^t, \, a' \neq a_k \,\}.$$

For attributes $a_k \in \mathcal{A}_{\text{high}}^t$, we align segment embeddings with contextualized value representations, promoting instance-level discrimination within each table:

$$\mathcal{L}_{\text{high}}^t = \mathbb{E}_{r \sim \mathcal{B}^t, \, a_k \in \mathcal{A}_{\text{high}}^t}\Big[\mathcal{L}_{\text{NCE}}\big(\sigma_{r,k}, \, \mathbf{e}_{r,k}^{(v)}, \, \mathcal{E}_{\text{high}}^-, \, \tau_{\text{high}}\big)\Big],$$
$$\mathcal{E}_{\text{high}}^- = \{\, \mathbf{e}_{r',k}^{(v)} \mid r' \in \mathcal{B}^t, \, r' \neq r \,\}.$$

Finally, given a batch $\mathcal{B}$ and a balancing hyperparameter $\lambda_{\text{align}}$, the full training objective is:

$$\mathcal{L}_{\text{total}} = \mathcal{L}_{\text{msm}} + \lambda_{\text{align}} \cdot \mathcal{L}_{\text{align}},$$

where $\mathcal{L}_{\text{align}} = \frac{1}{|\mathcal{T}_{\mathcal{B}}|} \cdot \sum_{t \in \mathcal{T}_{\mathcal{B}}} \big(\mathcal{L}_{\text{low}}^t + \mathcal{L}_{\text{high}}^t\big)$. Appendix A provides a theoretical analysis of the contrastive alignment.

## 4. Experiments

In this section, we evaluate whether NAVI effectively captures attribute semantics from both schema-level structural evidence and column-level distributional evidence across heterogeneous tables. We first assess schema-context modeling through generative reconstruction tasks, measuring the model's ability to capture functional dependencies between headers and values within rows. Next, we evaluate semantic consistency under lexical and structural variation, examining whether representations remain stable across heterogeneous header realizations while preserving domain-specialized attribute semantics. Finally, we evaluate downstream utility through row-level separability across diverse classifiers and the framework's integration as a robust semantic encoder within traditional feature-engineering pipelines.

### 4.1. Experimental Setup

**Datasets.** We evaluate on two domains, Movie and Product, using subsets of WDC WebTables (Peeters et al., 2024), selecting the 100 largest tables per domain. To balance domains, we downsample Product tables to a fixed target of 480,817 total rows via proportional (per-table) sampling. All tables are processed through a standardized pipeline (see Appendix D.2) to ensure compatibility with models that use BERT-style tokenization, and we split rows into train/validation/test with an 8:1:1 ratio. We utilize held-out subsets (10% test split) per domain. We uniformly subsample 1,000 rows per each evaluation run.

*Table 1.* Performance on header prediction and value imputation.

| Model | Product | | Movie | |
|---|---|---|---|---|
| | Header | Value | Header | Value |
| BERT | 0.9400 | 0.7743 | 0.9233 | 0.6889 |
| TAPAS | 0.2996 | 0.2501 | 0.2619 | 0.2326 |
| HAETAE | 0.9260 | 0.7785 | 0.8982 | 0.6911 |
| **NAVI** | **0.9995** | 0.7933 | **0.9998** | **0.7077** |
| w/o CAE | 0.5996 | 0.5544 | 0.5485 | 0.4799 |
| w/o MSM | 0.9959 | 0.7769 | 0.9941 | 0.6792 |
| w/o ESA | 0.9993 | **0.7977** | 0.9994 | 0.7012 |

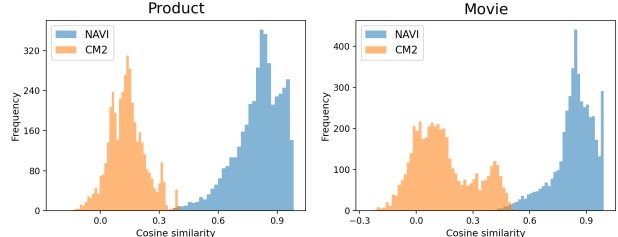

*Figure 3.* Distribution of cosine similarities between masked field representations and corresponding target representations.

**Baseline Methods.** We evaluate NAVI against representative table embedding models spanning major paradigms. BERT serves as the generic transformer backbone underlying most language model-based table encoders; its performance highlights the limitations of applying vanilla language models to tabular data. TAPAS, the most widely adopted table encoder, exemplifies structure-aware approaches, while CM2 and HAETAE represent feature-oriented and domain-aware encoders, respectively. This selection enables a systematic comparison of relevant work.

**Implementation.** We configure NAVI to balance domain and structural objectives. For alignment, we set $\tau_{\text{low}}, \tau_{\text{high}}{=}0.1, 0.02$, $\theta_{\text{low}}, \theta_{\text{high}}{=}5\%, 95\%$, and $\lambda_{\text{align}}{=}0.0125$ by default. For masked segment modeling, we use the ratio of header-masked, value-masked, vanilla MLM segments is 4:4:2. All models are trained for 2 epochs (the point at which learning curves indicate empirical convergence) with a batch size of 32 using AdamW (Loshchilov & Hutter) with a learning rate of $3 \times 10^{-5}$, and a weight decay of 0.01. Additional implementation details appear in Appendix D.

### 4.2. Generative Task Performance

To evaluate schema-context modeling, we assess whether NAVI can recover masked structural and semantic information from header–value dependencies within rows. Specifically, we consider two generative reconstruction tasks: *Header Prediction*, which requires recovering masked headers from associated values and surrounding schema context, and *Value Imputation*, which infers masked values conditioned on headers and neighboring row context.

As shown in Table 1, NAVI achieves near-perfect header prediction performance across both domains, substantially outperforming existing baselines. This result indicates that the proposed schema-aware contextualization successfully preserves stable attribute semantics even under heterogeneous header realizations. Furthermore, NAVI consistently achieves the best value imputation accuracy, suggesting that masked segment modeling effectively captures structural dependencies between headers and values within rows.

While the results in Table 1 evaluate exact token reconstruction, feature-oriented methods such as CM2 operate under a fundamentally different objective by reconstructing continuous feature embeddings instead of discrete subword tokens. To provide a more comparable evaluation across modeling paradigms, we further analyze semantic reconstruction quality in latent representation space.

For NAVI, we construct semantic field representations by mean-pooling contextualized hidden states over all subword tokens belonging to a header and its value. We then compute the cosine similarity between representations obtained from a masked forward pass and those from the corresponding unmasked gold forward pass. This measures whether the model preserves the semantic identity of masked fields beyond exact token recovery. Figure 3 visualizes the resulting similarity distributions for NAVI and CM2.

The distributions exhibit a clear separation between the two approaches across both domains. Notably, the lower quartile of NAVI already exceeds the upper quartile of CM2 (Product: $Q1_{\text{NAVI}} = 0.7400$ vs. $Q3_{\text{CM2}} = 0.1903$; Movie: $0.7870$ vs. $0.2578$), indicating that even weaker reconstructions produced by NAVI remain more semantically aligned with the target representations than the strongest reconstructions of CM2. These findings suggest that the proposed segment-centric schema modeling preserves substantially richer semantic information during masked reconstruction.

### 4.3. Consistency of Header Representations

To evaluate semantic consistency under heterogeneous schemas, we probe whether NAVI can consolidate disparate surface forms into a coherent domain context—a prerequisite for robust reasoning across unaligned tables. Specifically, we assess the model's ability to transcend superficial lexical cues and recognize semantically equivalent attributes despite the *lexical and statistical divergence* prevalent in real-world tables. Evaluation encompasses two distinct dimensions: first, the semantic consolidation of heterogeneous headers into stable tables, and second, the representational resilience of these schema anchors when reconstructing attributes from perturbed inputs and localized row context.

*Table 2.* Robustness of header prediction under schema perturbations. We evaluate the model's ability to reconstruct masked headers using perturbed schemas and their remaining context as reference. Accuracy is reported for the default test set and three perturbation settings: column permutation, synonym replacement, and typographical errors. For each perturbation, the highest accuracy is shown in bold, while the smallest relative performance drop from the default setting is highlighted in bold within parentheses.

| | **Product** | | | | **Movie** | | | |
| | Default | Col. Perm. | Synonym | Typo | Default | Col. Perm. | Synonym | Typo |
|---|---|---|---|---|---|---|---|---|
| BERT | .9400 | .8730 (-7.12%) | .9189 (-2.24%) | .8855 (-5.80%) | .9233 | .8590(-6.96%) | .9044 (-2.05%) | .8762 (-5.10%) |
| TAPAS | .2996 | .2681 (-10.51%) | .2457 (-17.99%) | .2539 (-15.25%) | .2619 | .2331 (-11.00%) | .2472 (-5.61%) | .2328 (-11.11%) |
| HAETAE | .9260 | .8619 (-6.92%) | .9146 (**-1.23%**) | .8800 (**-4.97%**) | .8982 | .8189 (-8.83%) | .8889 (**-1.04%**) | .8504 (-5.32%) |
| **NAVI** | .9995 | **.9997** (**+0.02%**) | **.9760** (-2.35%) | **.9375** (-6.20%) | .9998 | **.9997** (**-0.01%**) | **.9870** (-1.28%) | **.9522** (**-4.76%**) |

*Table 3.* Performance on header clustering. NMI and B³-F1 scores for clustering (using Agglomerative).

| Model | **Product** | | **Movie** | |
| | NMI | B³-F1 | NMI | B³-F1 |
|---|---|---|---|---|
| BERT | 0.8749 | 0.7317 | 0.8798 | 0.6969 |
| TAPAS | 0.8750 | 0.7239 | 0.8726 | 0.6759 |
| HAETAE | 0.8742 | 0.7268 | 0.8864 | 0.7276 |
| **NAVI** | **0.9005** | **0.7920** | **0.9144** | **0.7996** |

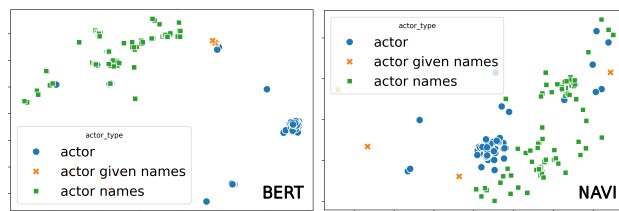

*Figure 4.* UMAP projection of contextualized header embeddings.

**Header Clustering.** To evaluate semantic consistency under lexical diversity, we cluster semantically equivalent headers across tables using agglomerative clustering, measured by B³-F1 and NMI. A model with consistent schema representations should group lexical variants of the same concept into compact, well-separated clusters. As shown in Table 3, NAVI substantially outperforms all baselines across both Product and Movie domains, indicating more reliable alignment of semantically equivalent headers despite variations in surface form.

To further analyze the geometry of the learned representation space, we measure pairwise distances between lexical variants (e.g., `actor` vs. `actor names`) and evaluate cluster separability using the silhouette score. Compared to BERT, NAVI produces smaller inter-variant distances under both cosine similarity (0.30 vs. 0.36) and L2 distance (0.72 vs. 0.77), indicating stronger semantic consolidation of lexical aliases. At the same time, NAVI preserves clearer separation between distinct semantic groups (e.g., `actor` vs. `director`), achieving a substantially higher silhouette score (0.52 vs. 0.26). These results suggest that NAVI improves both intra-semantic alignment and inter-semantic separation without collapsing distinct attribute semantics.

Figure 4 provides a qualitative illustration of this effect using the `actor` header group. Under NAVI, lexical variants collapse into a coherent semantic cluster despite differences in surface realization. In contrast, BERT produces fragmented clusters that remain separated according to lexical form. This contrast indicates that BERT primarily encodes

surface-level similarity, whereas NAVI consolidates lexical aliases into stable schema-level semantic representations. Together, these findings demonstrate that entropy-driven segment alignment effectively enforces inter-table semantic consistency under heterogeneous schemas.

**Robustness to Schema Perturbations.** We further evaluate the consistency of header representations by testing the model's resilience to controlled perturbations that reflect heterogeneity observed in real-world table corpora. We apply these perturbations only at inference time, using models trained strictly on clean schemas. Specifically, we consider three perturbation types. First, *column permutation* involves randomly shuffling the column order within each row while maintaining header–value alignment. Second, we apply *lexical perturbations* to 50% of low-entropy headers by either performing *synonym replacement* with unseen alternatives or introducing *header typos* through 1–2 character edits.

Table 2 reveals that while both NAVI and HAETAE provide baseline stability against lexical noise, they diverge significantly under structural drift. Against synonym and typographical perturbations, the performance gap is marginal, as both models leverage schema-level anchors to resist lexical drift. However, NAVI proves far more robust to structural variation; while HAETAE's accuracy drops up to 9% under column permutations, NAVI remains nearly invariant. This distinction confirms that strictly schema-aware approaches still rely on table-local structural cues, whereas NAVI's entropy-driven value alignment successfully decouples semantic roles from fragile structural artifacts.

*Table 4.* Performance on discriminative task. We evaluate row classification performance using Macro-F1 scores across the Product and Movie datasets, employing four diverse classifiers (XGBoost, Logistic Regression (LR), and TabPFN) operating on either [CLS] token embeddings or raw features. Methods are categorized into LM Encoders and FE Pipelines. For each classifier-category pair, the best result is indicated in bold, while the second-best is underlined. The overall best performance across all methods for a given classifier and dataset is marked with a dagger (†).

| | | Product | | | Movie | | |
|---|---|---|---|---|---|---|---|
| **Category** | **Model** | XGBoost | LR | TabPFN | XGBoost | LR | TabPFN |
| LM Encoders | BERT | 0.912 (± 0.013) | 0.931 (± 0.015) | 0.937 (± 0.020) | 0.595 (± 0.036) | 0.638 (± 0.043) | 0.659 (± 0.038) |
| | TAPAS | 0.911 (± 0.013) | 0.925 (± 0.015) | 0.930 (± 0.013) | **0.629 (± 0.035)**† | **0.669 (± 0.023)**† | **0.688 (± 0.017)**† |
| | CM2 | 0.814 (± 0.023) | 0.846 (± 0.018) | 0.842 (± 0.017) | 0.379 (± 0.025) | 0.375 (± 0.020) | 0.402 (± 0.015) |
| | HAETAE | 0.915 (± 0.015) | 0.938 (± 0.011) | 0.933 (± 0.013) | 0.602 (± 0.050) | 0.644 (± 0.041) | 0.664 (± 0.016) |
| | **NAVI** | **0.928 (± 0.010)** | **0.939 (± 0.012)** | **0.946 (± 0.013)**† | 0.615 (± 0.027) | 0.667 (± 0.028) | 0.679 (± 0.030) |
| | w/o CAE | 0.725 (± 0.025) | 0.854 (± 0.026) | 0.801 (± 0.019) | 0.265 (± 0.027) | 0.381 (± 0.028) | 0.303 (± 0.021) |
| | w/o MSM | 0.845 (± 0.024) | 0.903 (± 0.012) | 0.894 (± 0.015) | 0.467 (± 0.025) | 0.560 (± 0.027) | 0.522 (± 0.037) |
| | w/o ESA | 0.909 (± 0.019) | 0.930 (± 0.012) | 0.934 (± 0.013) | 0.623 (± 0.010) | 0.667 (± 0.010) | 0.674 (± 0.010) |
| FE Pipelines | Raw | 0.887 (± 0.015) | 0.829 (± 0.013) | 0.910 (± 0.013) | 0.457 (± 0.038) | 0.420 (± 0.035) | 0.478 (± 0.019) |
| | TableVec. | 0.920 (± 0.017) | 0.915 (± 0.014) | 0.931 (± 0.019) | 0.622 (± 0.014) | 0.552 (± 0.029) | N/A |
| | **NAVI**$_{FE}$ | **0.944 (± 0.017)**† | **0.942 (± 0.012)**† | **0.942 (± 0.015)** | **0.629 (± 0.027)** | **0.661 (± 0.028)** | **0.665 (± 0.031)** |

## 4.4. Downstream Discriminative task

To evaluate downstream utility, we assess whether semantically consistent representations learned from heterogeneous tables improve row-level discriminative performance. For LM-based encoders, each row is serialized and encoded once using the final [CLS] embedding as the fixed feature vector. We report macro-F1 on 10 balanced classes per domain (top product categories; top movie genres) using XGBoost, Logistic Regression, and TabPFN, averaged over eight subsampled runs (1,000 rows/run).

As shown in Table 4, NAVI achieves the strongest or near-strongest performance across all classifiers in the Product domain, confirming that resolving semantic asymmetry via entropy-driven alignment improves entity-level separability. However, performance dynamics shift in the Movie domain, where the schema is characterized by high redundancy (e.g., *actor.1*, *actor.2*). In this specific context, TAPAS leverages its strong intra-table structural priors to achieve the highest scores. Conversely, the Product domain features a semantically more diverse and cleaner schema, allowing NAVI's ability to reconcile global anchors with instance context to yield more significant gains.

Beyond LM-based encoders, we evaluate feature-engineering pipelines on the same task, including raw-feature baselines and TableVectorizer. While TableVectorizer performs type-aware preprocessing, it produces very high-dimensional representations in text-heavy domains such as Movie (>2,000 dimensions), making it incompatible with TabPFN and substantially larger than NAVI's 768-dimensional embeddings, yet without outperforming NAVI. To illustrate the benefit of replacing generic text encoders, we construct a simple hybrid model, NAVI$_{FE}$,

which concatenates raw numerical features with NAVI's text-derived segments. This hybrid already outperforms both raw-feature and TableVectorizer baselines, indicating that NAVI can serve as an effective drop-in semantic encoder within feature-engineering pipelines.

## 4.5. Ablation Study

We conduct ablation studies to analyze the contributions of NAVI from three perspectives: (1) the effectiveness of its core modeling components, (2) the role of entropy-based categorization in entropy-driven segment alignment, and (3) the influence of numerical features within the dataset.

**Main Components.** We ablate the core components of NAVI to evaluate their individual contributions to schema-context modeling and cross-table semantic consistency. One component is removed at a time while keeping all other settings fixed. We evaluate three variants: (1) w/o Context-free Attribute Embedding (CAE), removing the injection of transferable attribute priors into token and segment representations; (2) w/o Masked Segment Modeling (MSM), replacing structured segment masking with vanilla MLM; and (3) w/o Entropy-driven Segment Alignment (ESA), removing entropy-driven contrastive alignment across segments.

As shown in Table 1, removing CAE and MSM causes substantial degradation on generative reconstruction tasks, indicating that transferable attribute priors and segment-aware masking are both essential for modeling header–value dependencies. In Table 4, removing ESA substantially reduces Product performance, while its effect is weaker in Movie. This suggests that ESA is most beneficial when schemas are semantically diverse and require cross-table alignment, whereas Movie contains more redundant schema patterns.

*Table 5.* Effect of entropy-based routing in ESA. Imp denotes value imputation accuracy, and Cls denotes classification performance using XGBoost. Classification results are reported over five runs.

|  | Product | | Movie | |
|---|---|---|---|---|
|  | Imp | Cls | Imp | Cls |
| NAVI | **0.7933** | **0.928±0.010** | **0.7077** | **0.615±0.021** |
| RR | 0.7903 | 0.902±0.022 | 0.7071 | 0.592±0.022 |

**Entropy-based Categorization.** To evaluate whether ESA benefits specifically from entropy-based categorization, we compare it with a random routing (RR) variant that assigns alignment objectives to randomly selected columns. As shown in Table 5, RR consistently reduces row classification performance while producing smaller but consistent drops on value imputation. These results suggest that entropy-based routing provides an effective inductive bias by distinguishing stable domain attributes from highly instance-specific attributes during alignment.

**Influence of Numerical Features.** Unlike feature-oriented table encoders that explicitly model numerical feature types, NAVI primarily focuses on refining domain-specialized semantics from textual evidence. To evaluate whether its performance nevertheless depends heavily on numerical regularities, we analyze three settings on row classification with XGBoost: (1) *All*, using the full tables; (2) *Text-only*, removing numerical columns; and (3) *Partial-text*, reducing textual coverage while preserving table structure.

Across five runs, NAVI achieves $0.928 \pm 0.010$ under the full setting, $0.902 \pm 0.014$ under the text-only setting, and $0.888 \pm 0.015$ under the partial-text setting. The relatively small degradation after removing numerical columns indicates that performance is not primarily driven by numerical regularities. In contrast, reducing textual semantic evidence causes noticeably larger degradation, suggesting that NAVI mainly derives its gains from modeling textual semantics rather than specialized numerical feature encoding.

### 4.6. Geometry of Segment Embeddings

To empirically validate the advantages of Entropy-driven Segment Alignment, we analyze the representational geometry of segments from BERT and NAVI trained on the Movie domain. This visualization serves as a direct probe into how our alignment objective resolves the tension between schema stability and local instance discriminability. For BERT, segment embeddings are obtained by mean-pooling contextualized token embeddings over each header–value span, and all embeddings are projected with t-SNE.

As shown in Figure 5, the segment geometry of BERT reveals the absence of explicit header–value alignment. While high- and low-entropy segments are roughly separable, low-

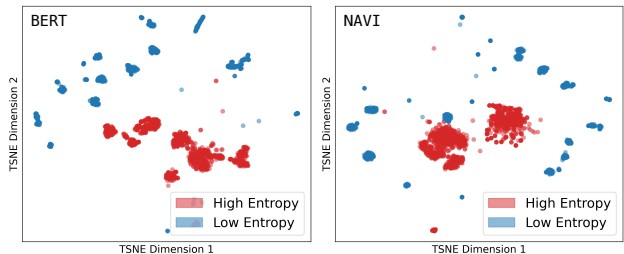

*Figure 5.* T-SNE visualization of segment embeddings from five heterogeneous Movie tables.

entropy segments (intended as stable anchors) are widely scattered, reflecting the entanglement of schema semantics with row-specific noise. High-entropy segments further fragment into table-specific micro-clusters, indicating that similarity is driven by superficial artifacts rather than consistent entity semantics. This geometry reflects the lack of role-aware organization in contextual embeddings learned by conventional LM-based encoders.

In contrast, NAVI exhibits a structured geometry induced by entropy-driven segment alignment. Low-entropy segments collapse into compact, well-localized clusters that act as domain anchors, reflecting cross-header alignment around global semantic centroids. This contraction shows that NAVI consistently extracts schema-level semantics while suppressing table-specific variation. High-entropy segments form broader, dispersed clusters that preserve row separability within the domain. This selective geometry, contracting anchors while maintaining instance variability, confirms that NAVI successfully decouples stable domain logic from entity-specific content.

## 5. Conclusion

In this paper, we address the longstanding tension between domain-level semantic consistency and local instance specificity by introducing the header–value segment as the atomic unit for tabular representation learning. Through Masked Segment Modeling and Entropy-driven Segment Alignment, NAVI jointly models schema-level structural context and column-level distributional context, yielding near-perfect generative schema reconstruction and strong resilience to structural and lexical drift. Qualitatively, the resulting embedding space exhibits a structured geometry where low-entropy anchors provide global stability while high-entropy segments maintain instance-level discriminability. Furthermore, the superior performance of NAVI_FE positions our framework as a potent drop-in semantic encoder for traditional feature-engineering pipelines. By bridging the gap between symbolic tabular structures and contextualized neural representations, this work establishes a robust foundation for future work in cross-domain transfer and the integration of unaligned, heterogeneous table corpora.

## Acknowledgments

This work was partly supported by the Institute of Information & Communications Technology Planning & Evaluation (IITP)-ICT Creative Consilience Program (IITP-2026-RS-2020-II201819), Artificial Intelligence Star Fellowship Support Program (IITP-2026-RS-2025-02304828), and the National Research Foundation of Korea (NRF) (RS-2024-00406320 and RS-2026-25494369) funded by the Korea government(MSIT).

## Impact Statement

This paper presents a framework for learning semantically consistent representations from heterogeneous in-domain tables. We believe this work can improve structured-data understanding and enable future applications involving large language model and table interactions, including table question answering, retrieval-augmented generation, and enterprise knowledge retrieval over heterogeneous tables. While our work is primarily methodological, its broader impact depends on how such representation systems are deployed and integrated into downstream applications.

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

# A. Theoretical Analysis of Entropy-driven Segment Alignment

We provide a geometric analysis of Entropy-driven Segment Alignment (ESA), the contrastive objective introduced in Section 3.3. Our goal is to analyze and characterize the local geometric forces induced by ESA on normalized segment embeddings, rather than to establish a generalization guarantee. Following the alignment–uniformity perspective of contrastive representation learning (Wang & Isola, 2020), we interpret contrastive objectives as simultaneously pulling positive pairs together and pushing negative pairs apart. ESA specializes this principle to heterogeneous tables by conditioning the choice of positives and negatives on the empirical entropy of attributes.

Unlike conventional contrastive learning, which applies a uniform positive-pair construction to all representations, ESA routes segment embeddings according to the semantic role of their associated attributes. For low-entropy attributes, segments are aligned with context-free attribute embeddings, encouraging stable schema-level anchors shared across rows and tables. For high-entropy attributes, segments are aligned with row-specific value representations while contrasted against other rows from the same table, preserving instance-level separability. This routing induces two complementary geometric effects: low-entropy segments undergo anchor contraction, whereas high-entropy segments maintain controlled dispersion.

## A.1. Setup and Notation

We analyze ESA in the segment embedding space. Let $\sigma_{r,k}$ denote the segment embedding for the $k$-th header–value realization in row $r$, as defined in Section 3.3. Let $\mathbf{e}_{a_k}$ be the context-free embedding of the associated domain attribute $a_k$, and let $\mathbf{e}_{r,k}^{(v)}$ be the contextualized value representation of value $v_{r,k}$. Throughout this analysis, we assume that all embeddings are $\ell_2$-normalized and lie on the unit hypersphere $\mathbb{S}^{d-1} = \{u \in \mathbb{R}^d : \|u\|_2 = 1\}$:

$$\sigma_{r,k}, \ \mathbf{e}_{a_k}, \ \mathbf{e}_{r,k}^{(v)} \in \mathbb{S}^{d-1}. \tag{2}$$

Accordingly, similarity is measured by the inner product, $\mathrm{sim}(u, v) = u^\top v$, which corresponds to cosine similarity on the unit hypersphere. We also use the standard identity

$$\|u - v\|_2^2 = 2(1 - u^\top v), \quad u, v \in \mathbb{S}^{d-1}, \tag{3}$$

which connects the dot-product form of InfoNCE to Euclidean geometry on normalized embeddings.

For a table $t$, ESA partitions its observed attributes into low-entropy and high-entropy subsets, denoted by $\mathcal{A}_{\mathrm{low}}^t$ and $\mathcal{A}_{\mathrm{high}}^t$, respectively. The entropy partition is computed from empirical column-value distributions before training and is treated as fixed during optimization.

For a low-entropy attribute $a_k \in \mathcal{A}_{\mathrm{low}}^t$, ESA uses the context-free attribute embedding $\mathbf{e}_{a_k}$ as the positive target for the segment embedding $\sigma_{r,k}$. The negatives are the context-free embeddings of other low-entropy attributes in the same table. Thus, the low-entropy loss for a segment is:

$$\ell_{\mathrm{low}}(\sigma_{r,k}) = -\log \frac{\exp(\sigma_{r,k}^\top \mathbf{e}_{a_k} / \tau_{\mathrm{low}})}{\exp(\sigma_{r,k}^\top \mathbf{e}_{a_k} / \tau_{\mathrm{low}}) + \sum_{a' \in \mathcal{A}_{\mathrm{low}}^t, \ a' \neq a_k} \exp(\sigma_{r,k}^\top \mathbf{e}_{a'} / \tau_{\mathrm{low}})}. \tag{4}$$

This objective defines positive pairs between segment embeddings and shared attribute-level anchors.

For a high-entropy attribute $a_k \in \mathcal{A}_{\mathrm{high}}^t$, ESA instead uses the row-specific contextualized value representation $\mathbf{e}_{r,k}^{(v)}$ as the positive target. The negatives are contextualized value representations of the same attribute from other rows in the same table. Let $\mathcal{B}^t$ denote the subset of batch rows sampled from table $t$. The high-entropy loss for a segment is:

$$\ell_{\mathrm{high}}(\sigma_{r,k}) = -\log \frac{\exp(\sigma_{r,k}^\top \mathbf{e}_{r,k}^{(v)} / \tau_{\mathrm{high}})}{\exp(\sigma_{r,k}^\top \mathbf{e}_{r,k}^{(v)} / \tau_{\mathrm{high}}) + \sum_{r' \in \mathcal{B}^t, \ r' \neq r} \exp(\sigma_{r,k}^\top \mathbf{e}_{r',k}^{(v)} / \tau_{\mathrm{high}})}. \tag{5}$$

This objective defines positive pairs between segment embeddings and row-specific value evidence, while using other rows as negatives.

This formulation highlights the key distinction of ESA: the same segment embedding $\sigma_{r,k}$ is regularized differently depending on the empirical entropy of its associated attribute. Low-entropy routing encourages segment embeddings to contract toward shared schema-level anchors, whereas high-entropy routing encourages them to remain distinguishable across rows. The following subsections formalize these two geometric effects.

**Lemma A.1** (Gradient of local InfoNCE). *Let $\sigma \in \mathbb{S}^{d-1}$ be a query representation, $x^+$ be its positive target, and $\mathcal{X}^- = \{x_j^-\}_{j=1}^M$ be a set of negative targets. Consider the local InfoNCE loss*

$$\ell(\sigma) = -\log \frac{\exp(\sigma^\top x^+/\tau)}{\exp(\sigma^\top x^+/\tau) + \sum_{j=1}^M \exp(\sigma^\top x_j^-/\tau)}. \tag{6}$$

*Let $\mathcal{X} = \{x^+\} \cup \mathcal{X}^-$, and define the softmax probability of any $x \in \mathcal{X}$ as*

$$p(x) = \frac{\exp(\sigma^\top x/\tau)}{\sum_{\tilde{x} \in \mathcal{X}} \exp(\sigma^\top \tilde{x}/\tau)}. \tag{7}$$

*Then the negative gradient of $\ell$ with respect to $\sigma$ is*

$$-\nabla_\sigma \ell = \frac{1}{\tau}\left[(1 - p(x^+))x^+ - \sum_{j=1}^M p(x_j^-)x_j^-\right]. \tag{8}$$

*Thus, the descent direction contains an attractive component toward the positive target and repulsive components away from the negative targets.*

*Proof.* Differentiating Eq. (6) with respect to $\sigma$ gives

$$\nabla_\sigma \ell = \frac{1}{\tau}\left(\sum_{x \in \mathcal{X}} p(x)x - x^+\right). \tag{9}$$

Taking the negative gradient yields

$$-\nabla_\sigma \ell = \frac{1}{\tau}\left(x^+ - \sum_{x \in \mathcal{X}} p(x)x\right). \tag{10}$$

Separating the positive and negative terms gives Eq. (8). □

### A.2. Low-entropy Alignment as Anchor Contraction

We first analyze the low-entropy branch of ESA. Low-entropy attributes typically correspond to stable, domain-coherent properties whose values exhibit limited variation within a table. For such attributes, ESA aligns each segment embedding $\sigma_{r,k}$ with its corresponding context-free attribute embedding $\mathbf{e}_{a_k}$. Since the same attribute embedding is shared across rows and tables, this objective encourages segment realizations of the same attribute to contract toward a common schema-level anchor.

**Theorem A.2** (Low-entropy ESA induces anchor-directed attraction). *Consider a low-entropy attribute $a_k \in \mathcal{A}_{\text{low}}^t$ and its segment embedding $\sigma_{r,k}$. Let $\ell_{\text{low}}(\sigma_{r,k})$ be the low-entropy InfoNCE loss defined in Eq. (4). Then*

$$-\nabla_{\sigma_{r,k}} \ell_{\text{low}} = \frac{1}{\tau_{\text{low}}}\left[(1 - p_{a_k})\mathbf{e}_{a_k} - \sum_{a' \in \mathcal{A}_{\text{low}}^t,\ a' \neq a_k} p_{a'}\mathbf{e}_{a'}\right], \tag{11}$$

*where $p_{a_k}$ is the softmax probability assigned to the positive anchor $\mathbf{e}_{a_k}$, and $p_{a'}$ is the probability assigned to a negative anchor $\mathbf{e}_{a'}$. Therefore, minimizing the low-entropy ESA objective moves $\sigma_{r,k}$ toward its attribute anchor while pushing it away from competing low-entropy attribute anchors.*

*Proof.* Apply Lemma A.1 with

$$\sigma = \sigma_{r,k}, \quad x^+ = \mathbf{e}_{a_k}, \quad \mathcal{X}^- = \{\mathbf{e}_{a'} \mid a' \in \mathcal{A}^t_{\text{low}}, \, a' \neq a_k\}, \quad \tau = \tau_{\text{low}}.$$

This gives Eq. (11). The interpretation follows from the attractive positive term and repulsive negative terms in the resulting descent direction.

$\square$

Theorem A.2 shows that the low-entropy branch of ESA imposes an anchor-based geometry on segment embeddings. Since all segments associated with the same low-entropy attribute share the positive target $\mathbf{e}_{a_k}$, their optimization directions contain a common attractive component toward the same schema-level anchor. This reduces variation caused by row-specific values and table-specific surface forms, encouraging compact clusters around the corresponding attribute anchor.

At the same time, the negative terms in Eq. (11) prevent trivial collapse across different low-entropy attributes. Each low-entropy attribute defines its own local anchor, while competing anchors repel one another through the contrastive denominator. Thus, ESA encourages low-entropy segments to become compact within the same attribute and separated across distinct attributes. This provides an objective-level explanation for the low-entropy geometry observed in Figure 5: stable attributes form compact semantic clusters that are less sensitive to row-specific noise. In the alignment–uniformity view (Wang & Isola, 2020), this corresponds to an entropy-conditioned form of alignment between stable header–value realizations and domain-level attribute anchors.

### A.3. High-entropy Alignment as Instance Separation

We next analyze the high-entropy branch of ESA. High-entropy attributes typically correspond to instance-specific information whose values vary substantially across rows. For such attributes, collapsing all segment embeddings toward a shared attribute anchor would remove useful entity-level distinctions. ESA therefore uses the row-specific contextualized value representation $\mathbf{e}^{(v)}_{r,k}$ as the positive target and contrasts it against value representations from other rows in the same table. This objective encourages segment embeddings to preserve local instance separability.

**Theorem A.3** (High-entropy ESA induces value-mediated instance separation). *Consider a high-entropy attribute $a_k \in \mathcal{A}^t_{\text{high}}$ and its segment embedding $\sigma_{r,k}$. Let $\ell_{\text{high}}(\sigma_{r,k})$ be the high-entropy InfoNCE loss defined in Eq. (5). Then*

$$-\nabla_{\sigma_{r,k}} \ell_{\text{high}} = \frac{1}{\tau_{\text{high}}} \left[ (1 - p_r)\mathbf{e}^{(v)}_{r,k} - \sum_{r' \in \mathcal{B}^t, \, r' \neq r} p_{r'}\mathbf{e}^{(v)}_{r',k} \right], \tag{12}$$

*where $p_r$ is the softmax probability assigned to the positive value representation $\mathbf{e}^{(v)}_{r,k}$, and $p_{r'}$ is the probability assigned to a negative value representation $\mathbf{e}^{(v)}_{r',k}$. Therefore, minimizing the high-entropy ESA objective moves $\sigma_{r,k}$ toward its own row-specific value representation while pushing it away from value representations of other rows.*

*Proof.* Apply Lemma A.1 with

$$\sigma = \sigma_{r,k}, \quad x^+ = \mathbf{e}^{(v)}_{r,k}, \quad \mathcal{X}^- = \{\mathbf{e}^{(v)}_{r',k} \mid r' \in \mathcal{B}^t, \, r' \neq r\}, \quad \tau = \tau_{\text{high}}.$$

This gives Eq. (12). The interpretation follows from the attractive positive term and repulsive negative terms in the resulting descent direction.

$\square$

Theorem A.3 shows that the high-entropy branch of ESA imposes an instance-discriminative geometry on segment embeddings. Unlike the low-entropy case, where multiple rows share the same attribute-level positive target, each high-entropy segment is aligned with its own row-specific value representation. Thus, ESA avoids contracting all realizations of a high-entropy attribute toward a single centroid and instead preserves the information needed to distinguish rows that share the same schema position but differ in value content.

The negative terms in Eq. (12) further induce controlled dispersion among row-specific realizations. Since negatives are sampled from other rows associated with the same high-entropy attribute, segment embeddings are separated relative to the empirical value distribution within the table rather than pushed toward arbitrary points on the hypersphere. This is particularly important for titles, names, identifiers, and other entity-specific fields, where schema-level anchoring would erase discriminative information. In the alignment–uniformity perspective, this corresponds to a entropy-conditioned analogue of uniformity: high-entropy segments remain dispersed while being constrained by contextualized value evidence. This explains the geometry observed in Figure 5, where NAVI preserves row-level discriminability.

### A.4. Combined Geometry of ESA

The preceding analysis shows that ESA induces a role-aware geometry in the segment embedding space. Rather than applying a single contrastive structure to all attributes, ESA changes the alignment target according to the empirical entropy of the associated column. Low-entropy attributes are aligned with shared context-free attribute anchors, yielding anchor-directed attraction and schema-level contraction. High-entropy attributes are aligned with row-specific value representations, yielding value-mediated separation and preserving instance-level discriminability. Thus, ESA jointly encourages semantic consistency for stable attributes and discriminability for entity-specific attributes.

This can be viewed as an entropy-conditioned adaptation of the alignment–uniformity perspective of contrastive learning (Wang & Isola, 2020). Low-entropy routing emphasizes alignment to global attribute anchors, while high-entropy routing introduces a localized uniformity-like effect by contrasting row-specific value realizations within the same table. Importantly, this does not imply global uniformity over the entire hypersphere; rather, it induces controlled dispersion within high-entropy segments. This role-aware geometry explains why NAVI can contract stable schema anchors without collapsing entity-specific information, supporting our central motivation that column-level distributional evidence should be incorporated selectively according to the heterogeneous semantic roles of attributes.

## B. Further Analysis on the Geometry of Segment Embeddings

To further examine how NAVI organizes segment embeddings, we visualize segment embeddings from five Movie tables using t-SNE, grouping segment embeddings by their entropy category (low vs. high) and overlaying table-wise convex hulls (Figure 6). The resulting geometry provides qualitative evidence that NAVI realizes cross-table generalization. For BERT, segment embeddings are constructed by mean-pooling the contextualized token embeddings of each header–value span, ensuring a comparable segment representation across models.

*Figure 6.* t-SNE projections of header–value segment embeddings from five Movie tables, grouped by entropy category. Gray convex hulls correspond to individual tables. For low entropy segment embeddings, points are additionally labeled as Best Rating or Worst Rating.

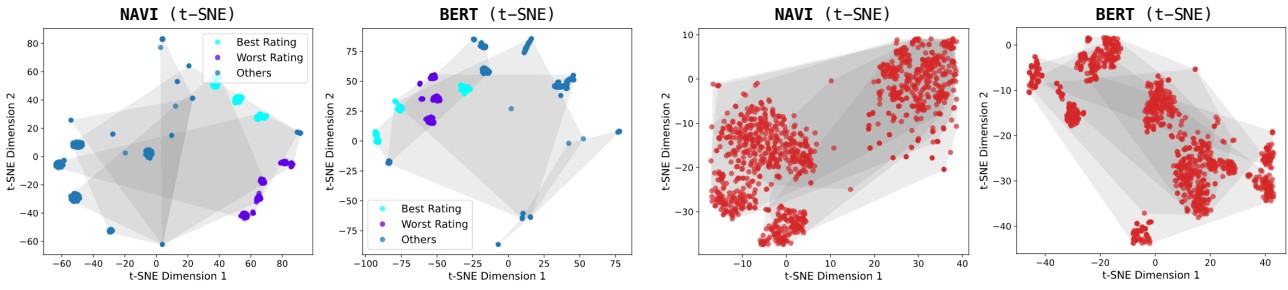

*(a)* Low entropy segment embeddings.        *(b)* High entropy segment distribution.

Low-entropy segment embeddings correspond to stable domain concepts (e.g., ratings, director). To assess whether models recover these semantics, we color segment embeddings using two coherent groups—Best Rating and Worst Rating. Under NAVI, the two groups form clean, well-separated clusters that persist across tables, with overlapping convex hulls indicating that the geometry is driven by shared cross-table value distributions rather than table identity. This reflects strong semantic discrimination and domain-level invariance. Overall, NAVI collapses surface-form variation while preserving core domain distinctions, yielding representations that generalize consistently across heterogeneous tables.

High-entropy segment embeddings represent entity-specific content (e.g., names, descriptions). NAVI distributes these segment embeddings broadly, avoiding collapse even when within-table contexts are similar. The table hulls heavily overlap, showing that representations do not cling to table identity. BERT, however, forms several dense, table-specific clumps, indicating that its contextual embedding remains sensitive to table-local patterns and fails to maintain row-level separability across tables. NAVI's geometry thus reflects that entity-specific values remain distinguishable without being entangled with schema or table-specific quirks.

# C. Additional Related Work: Predictive Tabular Learning

While the main paper focuses on table understanding and representation learning for heterogeneous tables, tabular data has also been studied extensively from the perspective of predictive modeling, generative modeling, and language-model-based table reasoning. We briefly discuss these broader directions and clarify how they differ from our goal of learning semantically consistent representations across heterogeneous in-domain tables.

Across these directions, prior work is largely organized around task-specific prediction objectives, such as predicting labels, missing values, textual answers, or executable programs. In contrast, our goal is not to optimize a single downstream task, but to learn reusable representations that resolve semantic inconsistency across heterogeneous in-domain tables, where latent attributes, their header realizations, and their value distributions may not be consistently aligned.

## C.1. Supervised Prediction on Tabular Features

A long line of work studies tabular learning as a supervised prediction problem, where the goal is to estimate target labels or continuous values from fixed input features. Classical machine learning methods, especially gradient-boosted decision trees (Chen & Guestrin, 2016; Ke et al., 2017; Prokhorenkova et al., 2018) remain strong baselines for classification and regression on structured tables. These methods are effective at modeling feature interactions, handling heterogeneous feature types, and achieving strong predictive performance under task-specific supervision. More recent neural approaches (Huang et al., 2020; Somepalli et al.; Gorishniy et al., 2021; Hollmann et al., 2023) further explore attention-based architectures, self-supervised objectives, and in-context prediction for tabular data.

These approaches typically assume fixed and well-defined feature spaces within a table or benchmark dataset. Our setting instead considers heterogeneous in-domain tables, where semantically related attributes may appear under different headers and identical headers may express different meanings depending on schema and column context.

## C.2. Generative Prediction for Imputation

Another line of work applies generative modeling to tabular data, with applications including data synthesis, missing value imputation, and distributional modeling. Recent methods adapt diffusion models, score-based models, and related generative objectives to structured tables (Kotelnikov et al., 2023; Kim et al.; Zhang et al., 2024a; Mueller et al.). These approaches model tabular feature distributions and can be used to generate realistic synthetic rows or recover missing entries (Zheng & Charoenphakdee, 2022). Related methods also explore conditional generation, mixed-type feature modeling, and robustness under incomplete observations.

These methods are related to our reconstruction-style objective, but they primarily model or complete tabular observations. In contrast, our masked segment modeling uses reconstruction to learn schema-aware header–value dependencies for representation learning.

## C.3. Language-mediated Prediction over Tables

With the emergence of large language models, recent work has also studied tables as inputs to language-centric reasoning systems. Earlier table pretraining methods formulate table reasoning as question answering, semantic parsing, or sequence-to-sequence generation over structured inputs (Herzig et al., 2020; Yin et al., 2020; Liu et al.). More recent systems adapt or instruction-tune large language models to interact with tables through natural language commands, table question answering, data analysis, or text-to-SQL-style reasoning (Zhao et al., 2024; Li et al., 2024; Zhang et al., 2024b). These methods are especially effective when tables are used as contexts for question answering, program generation, or symbolic reasoning.

These methods often treat tables as serialized prompts or external contexts for generating textual or symbolic outputs. Our work instead focuses on compact, reusable representations of heterogeneous tables themselves, which can complement LLM-table systems through retrieval, grounding, or preprocessing.

# D. Implementation Details

## D.1. Architectural Details

The context-free attribute encoder $f_{\theta'}$ introduced in Definition 3.2 is implemented as a lightweight BERT-based module that maps canonical attribute identifiers to context-independent attribute embeddings. The encoder uses the BERT tokenizer and embedding layer, followed by two Transformer layers selected from the pretrained BERT model, to leverage lexical and compositional priors for short attribute strings.

The use of a shallow encoder balances expressivity and efficiency. Since domain attributes are typically represented by short textual identifiers rather than full natural-language sentences, a lightweight encoder is sufficient to capture their lexical and compositional structure while avoiding unnecessary sentence-level contextualization. Using more layers may overfit to discourse-level semantics that are less relevant for attribute names, whereas using fewer layers limits the encoder's ability to model compositional attribute phrases.

We use layers 8 and 9 of the pretrained BERT encoder. This choice is motivated by prior analyses of BERT (Clark et al., 2019), which suggest that mid-to-deep layers capture syntactic and relational dependencies, while earlier layers focus more on local lexical information and final layers are more strongly shaped by sentence-level aggregation and task-specific adaptation. These properties make layers 8 and 9 suitable for modeling attribute identifiers, which are often short noun phrases requiring lexical and syntactic interpretation but not full discourse-level reasoning.

Given an attribute string $a$, the encoder first tokenizes it using the BERT tokenizer and obtains initial token embeddings through the BERT embedding layer:

$$\mathbf{u}^{(0)} = \mathrm{BertEmbeddings}(\mathbf{x}(a)).$$

The resulting sequence is then passed through two selected Transformer layers:

$$\mathbf{u}^{(1)} = \mathrm{EncoderLayer}_8(\mathbf{u}^{(0)}),$$
$$\mathbf{u}^{(2)} = \mathrm{EncoderLayer}_9(\mathbf{u}^{(1)}).$$

The context-free attribute embedding $\mathbf{e}_a$ is obtained by mean pooling over non-padding tokens:

$$\mathbf{e}_a = \frac{\sum_{i=1}^n \mathbf{u}_i^{(2)} \cdot \mathrm{mask}_i}{\sum_{i=1}^n \mathrm{mask}_i}.$$

This corresponds to the implementation of $f_{\theta'}$ and $\mathrm{Pool}(\cdot)$ in Definition 3.2.

The encoder supports single attribute strings, lists of attributes, and batched attribute lists, automatically adjusting the output dimensionality and applying attention masks for padded inputs.

**Projection Layer for Segment Embeddings.** The projection network $g(\cdot)$ used in the segment embedding definition in Section 3.3 combines three sources of evidence: the context-free attribute embedding $\mathbf{e}_{a_k}$, the contextualized header representation $\mathbf{e}_{r,k}^{(h)}$, and the contextualized value representation $\mathbf{e}_{r,k}^{(v)}$. These components correspond respectively to the domain-level attribute prior, the table-specific header realization, and the row-specific value realization of a segment.

Given a batch of segment components $\mathbf{E}_a \in \mathbb{R}^{B \times K \times D}$, $\mathbf{E}^{(h)} \in \mathbb{R}^{B \times K \times D}$, and $\mathbf{E}^{(v)} \in \mathbb{R}^{B \times K \times D}$, where $B$ is the batch size, $K$ is the number of segments per row, and $D$ is the hidden dimension, the projection first concatenates the three representations along the feature dimension:

$$\mathbf{z}_{r,k} = \mathbf{e}_{a_k} \parallel \mathbf{e}_{r,k}^{(h)} \parallel \mathbf{e}_{r,k}^{(v)} \in \mathbb{R}^{3D}.$$

The concatenated representation is then passed through a two-layer feedforward network with GELU activation, dropout, and layer normalization:

$$\mathbf{h}_{r,k} = \text{LayerNorm}\left(\text{GELU}\left(\text{Linear}_{3D \to 2D}(\mathbf{z}_{r,k})\right)\right),$$
$$\sigma_{r,k} = \text{LayerNorm}\left(\text{Linear}_{2D \to D}\left(\text{Dropout}(\mathbf{h}_{r,k})\right)\right).$$

The resulting vector $\sigma_{r,k}$ is the segment embedding used in Entropy-driven Segment Alignment.

This expansion–compression design follows the feedforward structure commonly used in Transformer architectures (Vaswani et al., 2017). The intermediate expansion allows nonlinear interactions among attribute-level, header-level, and value-level signals before projecting them back to the model dimension. Layer normalization (Ba et al., 2016) improves optimization stability, while the intermediate $2D$ representation provides sufficient capacity to capture dependencies among the components of each header–value segment.

### D.2. Dataset Preprocessing

Our dataset preprocessing pipeline is designed to optimize the quality and compatibility of tabular data for BERT-based language model training. The preprocessing consists of three main stages: data cleaning, BERT vocabulary validation, and tokenization optimization.

**Data Cleaning and Normalization**   The raw tabular data undergoes several cleaning steps to ensure consistency and quality. First, we flatten nested JSON structures.

```
"actors": [{"name": "allan"}, {"name": "daniel"}] →
"actors.0.name": "allan", "actors.1.name": "daniel"
```

This creates a uniform representation where each row is represented as a flat dictionary of key-value pairs. This flattening process preserves the hierarchical structure through dot-separated keys. Next, we handle indexed fields that represent repeated attributes. To prevent information overload and maintain computational efficiency, we sample a maximum of 3 indexed fields per field type, prioritizing the first occurrences to maintain data consistency.

**BERT Vocabulary Validation**   A critical challenge in training BERT on multilingual tabular data is the model's limited vocabulary coverage for non-English languages. To address this, we implement a BERT vocabulary validation step that filters out tables containing content that cannot be effectively tokenized by the BERT tokenizer.

For each table, we extract meaningful text fields (excluding URLs, pure numbers, and very short strings) and tokenize them using the BERT tokenizer. We calculate the ratio of unknown tokens ([UNK]) to total tokens for each field. Tables where more than 30% of the text fields contain excessive unknown tokens (threshold: 30% UNK ratio) are excluded from training. This filtering ensures that the model trains on data it can meaningfully process, significantly reducing the proportion of uninformative [UNK] tokens during training.

**Tokenization Optimization**   Finally, to maximize the utility of the remaining data while respecting BERT's token limit constraints, we implement field-level truncation: Individual fields that exceed 20 tokens are truncated to fit within this limit, preserving the most important information while maintaining field names and separators.

**Preprocessing Statistics**   Our preprocessing pipeline processes data from 100 different e-commerce websites across multiple languages and domains. The BERT vocabulary validation step typically filters out 60-70% of rows containing significant non-English content, resulting in a dataset focused on English-language e-commerce data that can be effectively processed by BERT.

The final preprocessed dataset maintains the structural information of the original tables while ensuring compatibility with BERT's tokenization scheme, enabling effective representation learning for tabular data through masked language modeling objectives.

This approach addresses the fundamental challenge of applying English-centric language models to multilingual structured data, ensuring that the training process focuses on content that the model can meaningfully learn from while preserving the rich structural information inherent in tabular data.

---

**Algorithm 1** NAVI Training Procedure

---

**Require:** Domain tables $\mathcal{T}$, model parameters $\theta$, alignment weight $\lambda_{\text{align}}$, masking configuration MaskCfg
**Ensure:** Trained parameters $\theta^*$
 1: Initialize model parameters $\theta$, optimizer, and gradient scaler
 2: **for** epoch $= 1, \ldots, T$ **do**
 3:     Construct table groups $\mathcal{G} = \{G_1, \ldots, G_M\}$ from $\mathcal{T}$
 4:     Shuffle group order
 5:     **for** each table group $G \in \mathcal{G}$ **do**
 6:         **for** each table $t \in G$ **do**
 7:             Compute the column entropy $\mathrm{H}_t(a)$ for each attribute $a \in \mathcal{A}^t$
 8:             Construct attribute partitions $\mathcal{A}^t_{\text{low}}$ and $\mathcal{A}^t_{\text{high}}$ using $\theta_{\text{low}}$ and $\theta_{\text{high}}$
 9:         **end for**
10:         Initialize a stratified sampler over rows from tables in $G$
11:         **for** each batch $\mathcal{B}$ sampled from $G$ **do**
12:             Serialize each row $r \in \mathcal{B}$ into segment sequence $\mathbf{x}(r)$
13:             Compute context-free attribute embeddings $\mathbf{e}_{a_k}$ using $f_{\theta'}$
14:             /* Masked Segment Modeling (Section 3.2) */
15:             Apply structure-aware masking using MaskCfg to obtain masked inputs and labels
16:             Forward masked inputs through $F_\theta$ to compute MSM logits
17:             Compute $\mathcal{L}_{\text{msm}}$ over masked token sets $\mathcal{M}_h$, $\mathcal{M}_v$, and $\mathcal{M}_r$
18:             /* Entropy-driven Segment Alignment (Section 3.3) */
19:             Forward unmasked inputs through $F_\theta$ to obtain contextualized token embeddings
20:             Extract $\mathbf{e}^{(h)}_{r,k}$ and $\mathbf{e}^{(v)}_{r,k}$ for each segment $\mathbf{s}_{r,k}$
21:             Construct segment embeddings $\sigma_{r,k} = g(\mathbf{e}_{a_k} \,\|\, \mathbf{e}^{(h)}_{r,k} \,\|\, \mathbf{e}^{(v)}_{r,k})$
22:             Compute $\mathcal{L}^t_{\text{low}}$ and $\mathcal{L}^t_{\text{high}}$ according to the source table $t$ of each row
23:             Compute $\mathcal{L}_{\text{align}} = \frac{1}{|\mathcal{T}_\mathcal{B}|} \sum_{t \in \mathcal{T}_\mathcal{B}} (\mathcal{L}^t_{\text{low}} + \mathcal{L}^t_{\text{high}})$
24:             Compute $\mathcal{L}_{\text{total}} = \mathcal{L}_{\text{msm}} + \lambda_{\text{align}} \mathcal{L}_{\text{align}}$
25:             Update $\theta$ using $\mathcal{L}_{\text{total}}$
26:         **end for**
27:     **end for**
28: **end for**
29: **return** $\mathcal{M}_{\theta^*}$

---

### D.3. Training Procedure

For each batch constructed as described in Section 3.3, the model performs two forward passes. The masked serialized rows are first passed through the table encoder $F_\theta$ to compute MSM logits and the masked segment modeling loss $\mathcal{L}_{\text{msm}}$ over $\mathcal{M}_h$, $\mathcal{M}_v$, and $\mathcal{M}_r$ defined in Section 3.2. The unmasked rows are then passed through $F_\theta$ to obtain contextualized token embeddings for entropy-driven segment alignment. We extract $\mathbf{e}^{(h)}_{r,k}$ and $\mathbf{e}^{(v)}_{r,k}$ following Definition 3.3, combine them with the context-free attribute embedding $\mathbf{e}_{a_k}$ from Definition 3.2, and compute the segment embedding $\sigma_{r,k} = g(\mathbf{e}_{a_k} \,\|\, \mathbf{e}^{(h)}_{r,k} \,\|\, \mathbf{e}^{(v)}_{r,k})$. These segment embeddings are used to compute $\mathcal{L}^t_{\text{low}}$ and $\mathcal{L}^t_{\text{high}}$ under the entropy-based attribute partitions in Section 3.3. The total loss follows the main objective: $\mathcal{L}_{\text{total}} = \mathcal{L}_{\text{msm}} + \lambda_{\text{align}} \mathcal{L}_{\text{align}}$.

