# OpenReview forum: "Segment-driven Structural Induction and Semantic Alignment for Heterogeneous Tabular Representation"
_ICML.cc/2026/Conference — ICML 2026 regular_

### Official Review · Reviewer_xtRM · 2026-03-06

**Soundness:** 2
**Presentation:** 1
**Significance:** 2
**Originality:** 2
**Overall Recommendation:** 4
**Confidence:** 3

**Summary:**

This work introduces the segment as the fundamental representation unit and utilizes the masked segment modeling and entropy-driven segment alignment to treat low-entropy segments as domain anchors and high-entropy segments as entity-specific signals. The comparison with baselins exhibit the strength of the proposed methods.

**Compliance With Llm Reviewing Policy:**

Affirmed.

**Final Justification:**

The authors have addressed my concerns, so I decided to increase the score. The proposed method NAVI has the largest model complexity (151M parameters) compared to baseline methods. The performance gain compared to baseline methods might be due to an increase in model complexity. Hence, I increase the score to "Weak accept".

**Key Questions For Authors:**

1. For tables that contain an equal or larger portion of numerical columns, will the performance of NAVI degrade?

If a table contains mostly numerical columns, it becomes semantically sparse. NAVI relies on the semantic understanding.

2. In the evaluation datasets Movie and Product, only the number of rows is reported. What is the number of columns?

Due to the context length limitation in the transformer-based models, a known issue of applying LLMs to the tabular data domain is the truncation of tables due to that limitation.

**Limitations:**

NAVI mainly focuses on the semantic understanding of tabular data. All evaluations are limited to text-heavy tabular domains.

**Strengths And Weaknesses:**

## Strengths

1. The entropy-based design is well motivated and supported by experimental results.

2. Illustration figure (Figure 2) uses concrete examples that are helpful for understanding the proposed method. Main contributions are clearly shown in the figure.

## Weaknesses

### Major Weaknesses

1. Lack of clarity in the presentation.

1.1. A segment is formally defined as a unit consisting of a context-free schema representation, a contextualized header and a value representation. On line 192 "for each token within the segment k ...", it indicates that the segment is a set of tokens instead of representations.

1.2. The equation on Line 197 indicates that the positional encoding is a function of the token. In that sense, how is the ordering of tokens enforced?

1.3. $E_{word}$ is used in that equation but not clearly explained. I guess it means word embedding. Please clearly define each term appearing in equations to increase readability. Similarly, $\mathcal{V}$ in the Equation on line 211 is not defined.

1.4. $g(\cdot)$ is explained on Line 214 as a projection. It is unclear whether it is a linear projection or a non-linear projection. It is also unclear whether this projection operation is trainable.

1.5. It is unclear to me for the words "Headers and values are semantically different". Does that simply say table header and cell have different values? The authors use this to explain why a direct application of MLM is sub-optimal. MLM can work well for sentences that contain different words.

2. The header/cell-dependent masking is not a new approach. It has been applied in many MLM pretraining (e.g. [1] and [2]). The claim in the manuscript is "To address this, we introduce...". The claim indicates that the header/cell-dependent masking is the innovation of this work.

3. Limited evaluation.

*Datasets*: Tabular datasets are limited to two areas: Movie and Product. It is hard to evaluate how NAVI performs in other areas, especially in areas where there is a fairly large amount of numerical columns.

*Baselines*: BERT is designed for general language understanding. TAPAS is designed for table parsing. Only HAETAE is designed for in-domain prediction. Please include SOTA baselines [3-5] to demonstrate the strength of NAVI.

4. The claim on "NAVI consolidates lexical aliases into coherent schema representation" is not supported by the experimental results.

In Figure 3, it is not visually obvious that NAVI collapses lexical variants into a single cluster better than BERT. Specifically, lexical variants under the actor class seem to collapse into a single cluster better in BERT compared to NAVI. Since the visualization is not obvious, the authors should use the metric to characterize the clustering.

### Minor Weaknesses

1. Please strictly follow the ICML rule to include Impact Statement as it is mandatory section.

2. What does the bold font mean in Table 3? Typically, the bold font is used to mark the best performance. Apparently, it is the purpose in Table 3.

3. In the performance comparison (Table 1), please report # trainable parameters and FLOPs for a comprehensive evaluation.



[1] Yin, Pengcheng, et al. "TaBERT: Pretraining for joint understanding of textual and tabular data." Proceedings of the 58th annual meeting of the association for computational linguistics. 2020.
[2] Yang, Yazheng, et al. "Unitabe: A universal pretraining protocol for tabular foundation model in data science." arXiv preprint arXiv:2307.09249 (2023).
[3] Ye, Chao, et al. "Towards cross-table masked pretraining for web data mining." Proceedings of the ACM Web Conference 2024. 2024.
[4] Yang, Yazheng, et al. "Unitabe: A universal pretraining protocol for tabular foundation model in data science." arXiv preprint arXiv:2307.09249 (2023).
[5] Zhang, Tianshu, et al. "Tablellama: Towards open large generalist models for tables." Proceedings of the 2024 Conference of the North American Chapter of the Association for Computational Linguistics: Human Language Technologies (Volume 1: Long Papers). 2024.

---

> ### Author Rebuttal · Authors · 2026-03-31
>
> We thank reviewer xtRM for the careful feedback and for highlighting the strengths of our entropy-based design. We believe that some parts of the presentation, particularly segment definition, notation, and header–value roles, were not clearly communicated, which may have caused confusion. In response, we provide (i) clarifications of notation and concepts, (ii) clearer distinction between 'segment' and 'segment embeddings', and refine our contribution beyond masking, and (iii) additional quantitative evidence and dataset analysis. Below, we address each concern.
>
>
> ### W1.1–W1.4 - Notation & Clarity
>
> `Segment Embedding` Adopting a notion of a segment as a set of tokens delimited by a saparator (e.g., [SEP]) in language modeling, we define a segment in the context of tabular modeling as a header–value pair, as formally introduced in Sec. 4. Thus, the phrase “for each token within the segment” refers to the tokenized header–value pair. The segment *embedding* is an instantiated representation of a segment that integrates a context-free schema embedding with contextualized header and value embeddings.
>
> `Terms in Equations.` $E_{\text{word}}$ denotes word embeddings, as you mentioned, $x^{(k)}_j$ is the $j$-th token in segment $k$, and $P_j$ is the positional embedding at position $j$, enforcing token ordering through standard positional encoding within each segment. $V$ denotes the *vocabulary*, and $g$ is a *trainable* two-layer feedforward network with GELU activation and layer normalization (Appendix D.1).
>
> We will revise the manuscript to explicitly define all terms and consistently distinguish between the segment and its corresponding segment embedding.
>
> ### W1.5, W2 - Conceptual Clarification
>
> `Functional Roles of Headers and Values.` Headers represent attributes and serve as semantic references, while values instantiate them with instance-specific information. Thus, headers encode schema-level semantics, whereas values encode context-dependent realizations. Standard MLM learns token co-occurrence patterns uniformly, but in tables the same value can have different semantics depending on its header. Therefore, modeling tabular data requires capturing header–value dependencies beyond co-occurrence.
>
> `Clarification of Our Contribution.` Masked segment modeling is a supporting objective for stabilizing segment representations. Our main contribution lies in (i) segment embeddings that integrate schema-level and instance-level signals for alignment, and (ii) entropy-driven segment alignment, which assigns different discriminative roles to columns and aligns segment embeddings accordingly. Together, these components enable modeling both schema-level consistency and instance-level variability.
>
> ### W3, W4, Q1, Q2 - Evaluation
>
> `Datasets.` We evaluate on two text-heavy domains (Movie, Product), each with **100 heterogeneous tables** from WDC WebTables. Our work achieves 0.928±0.010 (All), 0.902±0.014 (Text-only), and 0.888±0.015 (Partial-text), showing that removing numerical columns has limited impact while reducing text coverage causes larger degradation, indicating performance is primarily driven by textual semantics. Regarding table width, we define the number of columns as the number of header–value fields per row. Tables range from 5–408 (avg. 16) columns before preprocessing and 5–33 (avg. 13) after preprocessing, demonstrating that our pipeline effectively controls table width under context-length constraints (Appendix D.2).
>
> `Baselines.` We pick the current baselines that are reproducible and directly comparable in our probelm setting. While some recent methods lack publicly available implementations or are not fully aligned with our setting (e.g, a decoder-based TableLlama). We will include more relevant baselines (e.g., TABBIE), following your suggestion.
>
> `Header Clustering.` We quantify clustering behavior by measuring distances between lexical variants (e.g., actor vs. actor names), where our work reduces inter-variant distance compared to BERT (0.30 vs. 0.36 cosine; 0.72 vs. 0.77 L2), indicating stronger alignment of semantically equivalent headers. At the same time, separation between distinct semantic groups (e.g., actor vs. director) is better preserved, as shown by a substantially higher silhouette score (0.52 vs. 0.26). These results demonstrate that our work improves both intra-semantic alignment and inter-semantic separation, consolidating lexical aliases into coherent semantic representations without collapsing distinct semantics.
>
> ### Minor Weaknesses
> We appreciate your careful comments on these minor points.
> 1. We will include the Impact Statement in the revision.
> 2. Bold in Table 3 marks the smallest performance drop per perturbation.
> 3. Params: BERT 109M, TAPAS 110M, HAETAE 133M, Ours (NAVI) 151M.
>
> We hope our responses address your concerns and would be happy to discuss any further questions you may have.

---

> > ### Author Rebuttal · Reviewer_xtRM · 2026-03-31
> >
> > I thank the authors for addressing part of my concerns. Without showing the comparison with more recent baselines, it is difficult to show the strength of the proposed method.

---

> > > ### Author Response · Authors · 2026-04-07
> > >
> > > We thank the reviewer for the constructive suggestion to include more recent baselines. Following your kind recommendation, we have additionally integrated the open implementation of CM2 [1], the most recent baselines among the recommended references, into our empirical evaluation. We believe this addition significantly strengthens the comparison and provides a clearer picture of NAVI's performance against contemporary state-of-the-art methods. Since NAVI and other baselines are evaluated with only the checkpoint pretrained with MLM-based objectives, we evaluated CM2 with the checkpoint pretrained with pMTM (prompt Masked Table Modeling) objective mentioned in the paper.
> > >
> > > `Missing Value Imputation.` Models in our manuscript follow standard masked language modeling; we mask subword tokens in the cell text, and report exact token match accuracy. CM2 instead performs masked feature modeling at the column level: one vector per column, with an objective of cosine distance to mean-pooled BERT embeddings of cell text; evaluation of CM2 mirrors that with cosine similarity thresholds, not subword token identity. The two protocols therefore measure different targets (token IDs vs embedding targets) and different granularities (subwords vs columns). While a naive comparison using exact token match accuracy shows a massive disparity—with NAVI achieving 79.33%/70.77% accuracy on Product/Movie domain compared to CM2’s 6.92%/5.31%—this evaluation is inherently biased against the method of CM2.
> > >
> > > To rigorously assess value imputation performance, we move beyond discrete token matching and evaluate the models within their native latent spaces. For NAVI, we employ a dual-forward strategy: by mean-pooling the contextualized hidden states across all constituent subword indices within a field, we derive a single semantic representation that captures the model’s internal reconstruction of a cell’s identity. We then compute the cosine similarity between two such vectors: one generated from a fully masked forward pass and the other from a "gold" forward pass where all input tokens are restored. This protocol ensures that NAVI is evaluated on its ability to recover holistic semantic information, directly comparable to the feature-level evaluation of CM2.
> > >
> > > (Table 1: Value Imputation — distribution of cosine similarities between prediction, target)
> > > ||Prod-Q1|Prod-Q2|Prod-Q3|Mov-Q1|Mov-Q2|Mov-Q3|
> > > |-|-|-|-|-|-|-|
> > > |CM2|0.0716|0.1334|0.1903|0.0237|0.1172|0.2578|
> > > |NAVI|**0.5166**|**0.6489**|**0.7474**|**0.5051**|**0.6245**|**0.7625**|
> > >
> > > We report the quartiles (Q1, Q2, Q3) to provide a robust view of the semantic recovery distribution. The results demonstrate a profound disparity characterized by a complete quartile crossover: NAVI’s first quartile (Product: 0.5166; Movie: 0.5051) significantly exceeds the third quartile of the CM2 baseline (0.1903/0.2578). This indicates that the bottom 25% of NAVI’s semantic reconstructions outperform the top 25% of CM2’s specialized feature predictions, suggesting a systemic improvement in tabular understanding.
> > >
> > > `Row Classification.` We further evaluate the quality of the learned representations through row classification. By extracting the CLS embedding of a row and utilizing it as the input feature for standard ML-based classifiers, we can measure the discriminative power of each model’s latent space without the bias of specific prediction heads.
> > >
> > > (Table 2: Row Classification — F1 Score)
> > > ||Prod-XGB|Prod-CatB|Prod-LR|Mov-XGB|Mov-CatB|Mov-LR|
> > > |-|-|-|-|-|-|-|
> > > |CM2|0.814±0.023|0.805±0.013|0.846±0.018|0.379±0.025|0.370±0.015|0.375±0.020|
> > > |NAVI|**0.928±0.023**|**0.913±0.015**|**0.939±0.016**|**0.604±0.045**|**0.607±0.035**|**0.667±0.029**|
> > >
> > > NAVI’s segment-based representation learning achieves significantly higher F1 scores across all classifiers and domains than CM2. Specifically, in the Movie domain, NAVI nearly doubles the classification performance of CM2 (e.g., 0.667 vs. 0.375 for LR). These results demonstrate that NAVI’s consolidation of lexical aliases into coherent schema representations produces more robust, linearly separable features for downstream tasks than current column-level baselines.
> > >
> > > We particularly appreciate your suggestion to include CM2. We will ensure that these empirical results and the detailed architectural clarifications from our discussion are fully integrated into the final version. We hope this comprehensive comparison resolves your remaining questions and thank you again for the feedback that helped us significantly strengthen the paper.
> > >
> > > [1] Ye, Chao, et al. "Towards cross-table masked pretraining for web data mining." Proceedings of the ACM Web Conference 2024.

---

### Official Review · Reviewer_ofDh · 2026-03-13

**Soundness:** 3
**Presentation:** 3
**Significance:** 3
**Originality:** 3
**Overall Recommendation:** 4
**Confidence:** 4

**Summary:**

The paper proposes NAVI, a pretraining method for better representation on text-heavy tables using header-value “segments” as the atomic unit. NAVI encodes table headers and values and then trains with masked segment modeling and an entropy-based alignment loss, treating columns with low entropy as domain anchors and those with high entropy as entity-specific feature.  The paper benchmarked the method on the Movie and Product parts of WDC WebTables, showing NAVI outperform baselines table encoding methods(BERT, TAPAS, HEATAE) on header prediction, clustering, and is robuts to column shuffling and lexical perturbation. NAVI also show edge on some downstream prediction tasks on text table prediction.

**Compliance With Llm Reviewing Policy:**

Affirmed.

**Final Justification:**

During the discussion, the authors added the additional comparison against CM2 and the architectural discussion.

The empirical results showing NAVI's Q1 exceeding CM2's Q3 on value imputation, along with the row classification improvements, provide useful evidence that NAVI's representations are stronger than an existing cell-level baseline.

However, my core concern remains only partially addressed. the follow-up questions asked whether cell-level representations would fail under the same training objectives (MSM + ESA), which would isolate the segment formulation as the source of improvement. The provided comparison evaluates CM2 under its own native objective (pMTM), making it impossible to disentangle the contribution of the segment architecture from the contribution of the training recipe. The architectural rationale that the context-free anchor Hctx-free provides an invariant grounding signal absent in cell-level models is intuitive but remains a verbal argument rather than the formal or controlled empirical analysis expected. Without an ablation that applies ESA to cell-level embeddings (e.g., from TABBIE or TaBERT) and demonstrates degradation, the claim that the segment formulation is architecturally necessary rather than merely sufficient is not fully substantiated.

I therefore maintain my rating of weak accept.

**Key Questions For Authors:**

+ Is there any ablation study on the entropy-based routing effectiveness?

+ If you keep more of the multilingual or noisy rows instead of filtering them out, and will the method degrades gradually?

**Limitations:**

yes

**Strengths And Weaknesses:**

## Strengths

+ Good presentation and well motivated problem. The proposed header-value "segment" is a interesting and effective formulation that motivates the problem.  The main algorithm components are defined clearly.

+ Highly effective in various relevant tasks. NAVI outperformed classic


## Weaknesses

+ Missing cross table splits in training setting. The paper is framed as studying heterogeneous cross-table generalization, but the train/dev/test split is row-based within the same tables. That makes the setup feel weaker than the framing suggests.

+ The distinction/importance of segment (header-value pairs) deserves more discussion. Many classic prior works used the header-value pair, or cells, as basic unit of tabular modeling [1,2]. The paper will benefit from the better discussion and directly comparison between them. Current baseline set appears limited.

+ The experimental scope and gain also feels narrower than the generality claim. There are only two text-heavy domains, and a large share of non-English e-commerce rows is removed in preprocessing. The performance edge also diminished during downstream prediction task.

[1] Iida, Hiroshi, et al. "Tabbie: Pretrained representations of tabular data." Proceedings of the 2021 Conference of the North American Chapter of the Association for Computational Linguistics: Human Language Technologies. 2021.

[2] Yin, Pengcheng, et al. "TaBERT: Pretraining for joint understanding of textual and tabular data." Proceedings of the 58th annual meeting of the association for computational linguistics. 2020.

---

> ### Author Rebuttal · Authors · 2026-03-31
>
> We thank Reviewer ofDh for the positive assessment of our work, including the clear motivation, well-defined methodology, and especially appreciate the recognition of the segment formulation and the effectiveness of our framework. We also thank the reviewer for the constructive suggestions regarding cross-table generalization, the role of segments relative to prior work, and the scope of evaluation. In response, we address these points through (i) entropy-based routing ablations, (ii) evaluation scope, and (iii) the distinction from cell-level modeling.
>
> ### Q1 - Entropy-based Routing Ablations
> We evaluate the effectiveness of entropy-based routing in Entropy-driven Segment Alignment (ESA) by comparing NAVI (our work) with a random routing variant, where 10% of columns are randomly assigned to alignment objectives. We newly conduct five runs per experiment and report mean ± standard deviation.
>
> (Table 1)
> ||Prod(Imp)|Prod(Cls)|Mov(Imp)|Mov(Cls)|
> |-|-|-|-|-|
> |**NAVI** (Default)|0.786±0.009|0.928±0.010|0.708±0.008|0.604±0.021|
> |**NAVI** (Random Routing)|0.783±0.008|0.902±0.022|0.707±0.008|0.581±0.022|
>
> As shown in Table 1, entropy-based routing consistently improves performance. Replacing it with random routing leads to clear drops in classification, while imputation differences remain smaller but consistent. This indicates that entropy-based routing provides a meaningful inductive bias: by distinguishing domain-coherent columns from entity-specific ones, it guides alignment to better preserve discriminative structure. In contrast, random routing disrupts this separation, causing over-alignment of entity-specific attributes and under-alignment of domain anchors, which leads to degraded discriminative performance.
>
>
> ### W1, Q2, W3 - Discussion of Evaluation Scope
> `Cross-table Setting.` We would like to clarify that our notion of cross-table generalization differs from generalization to entirely unseen schemas. Our goal is to learn representations whose semantics are conditioned on schema context and empirical value distributions within a domain. In practice, table schemas are designed with domain-specific intent, and attribute meaning depends on both co-occurring attributes and value patterns rather than being defined in isolation. To capture this, we jointly train on multiple heterogeneous tables (100 per domain) so that the model can observe how semantically related attributes are instantiated across schema variations. This setting is essential to our objective: unlike prior context-aware encoders that operate on a single table and primarily capture intra-table dependencies, our model explicitly learns cross-table semantic consistency under schema variation. Accordingly, we adopt a row-based split to evaluate whether the model generalizes across instances drawn from this heterogeneous table collection, rather than relying on table-specific patterns. A table-level split would instead evaluate transfer to entirely unseen schemas, which we view as a complementary but distinct setting from our focus.
>
>
> `Multilingual, Noisy Rows.` Our preprocessing step filters out rows that cannot be reliably tokenized by the underlying language model (e.g., high [UNK] ratios due to multilingual or severely noisy text). In such cases, much of the input collapses into [UNK], making semantic evaluation unreliable. We instead assess robustness via controlled perturbations (Table 3 of the manuscript), including typographical noise, where NAVI remains stable. We agree that extending to multilingual settings is an important direction. However, it requires appropriate tokenization and is beyond our current scope; this limitation applies equally to all BERT-based baselines.
>
> We also note that downstream performance varies across domains (Table 4; Sec. 5.4): gains are more pronounced in Product, which exhibits higher schema heterogeneity, while methods with strong intra-table priors (e.g., TAPAS) perform better in more redundant schemas (e.g., actor.1, actor.2,...) such as Movie.
>
> ### W2 - Segments vs. Cell-level Modeling
> Prior cell-level modeling approaches (e.g., TaBERT, TABBIE) aim to accurately represent each cell by aggregating row- and column-wise context, focusing on intra-table understanding. In contrast, our segment is not intended as a stronger cell representation, but as a functional unit that mediates between schema-level and instance-level signals, enabling contrastive alignment between context-free header embeddings and contextualized value embeddings. This distinction shifts the focus from improving cell encoding to modeling cross-table semantic consistency under schema variation. Accordingly, we adopt TAPAS as a representative baseline for strong intra-table modeling.
>
> We thank you again for your valuable comments and will incorporate all the suggestions and feedback in the final manuscript.

---

> > ### Author Rebuttal · Reviewer_ofDh · 2026-04-03
> >
> > I appreciate the authors detailed responses. It is helpful to clarify the effectiveness of routing via ablation, as well as a more focused evaluation scope.
> >
> > The rebuttal argues that cell-level representations such as in TaBERT/TABBIE cannot serve as effective units for cross-table semantic alignment, and that the segment formulation is necessary for this purpose. However, the paper have not provided a theoretical argument nor an empirical comparison to support this claim. TaBERT and TABBIE can be seen as already encode header–value pairs through contextualized attention, so what fundamental architectural reasons specifically prevents their cell representations from being used as alignment units under the same training objectives (MSM + ESA)? Without a direct comparison or a formal analysis showing why cell embeddings are insufficient, the core novelty claim of the segment formulation remains not fully supported.

---

> > > ### Author Response · Authors · 2026-04-07
> > >
> > > We appreciate the reviewer’s insightful follow-up. To clarify why cell-level models (e.g., TaBERT, TABBIE) are insufficient as alignment units, we compare NAVI with CM2 [1], a recent state-of-the-art method in cell-level modeling, and provide the requested architectural rationale.
> > >
> > > ### Empirical Comparison
> > >
> > > NAVI and other baselines are evaluated with only the checkpoint pretrained with MLM-based objectives, thus we evaluate CM2 with the checkpoint pretrained with pMTM (prompt Masked Table Modeling) objective mentioned in the paper.
> > >
> > > `Missing Value Imputation.` Since CM2 optimizes for feature-level cosine distance while NAVI utilizes subword MLM, we evaluate both models within their native latent spaces to ensure a fair comparison. For NAVI, we employ a dual-forward strategy, computing the cosine similarity between the mean-pooled representation of a masked cell and its "gold" (unmasked) counterpart.
> > >
> > > Table 1 shows a complete quartile crossover: NAVI’s first quartile exceeds CM2’s third quartile in both domains. This indicates the bottom 25% of NAVI’s semantic reconstructions outperform the top 25% of CM2’s specialized feature predictions, suggesting a systemic improvement in tabular understanding.
> > >
> > > (Table 1: Value Imputation — distribution of cosine similarities between prediction, target)
> > > ||Prod-Q1|Prod-Q2|Prod-Q3|Mov-Q1|Mov-Q2|Mov-Q3|
> > > |-|-|-|-|-|-|-|
> > > |CM2|0.0716|0.1334|0.1903|0.0237|0.1172|0.2578|
> > > |NAVI|**0.5166**|**0.6489**|**0.7474**|**0.5051**|**0.6245**|**0.7625**|
> > >
> > > `Row Classification.` We further evaluate the quality of the learned representations through row classification by extracting the CLS embedding of a row and utilizing it as the input feature for standard ML-based classifiers. As shown in Table 2, NAVI demonstrates superior discriminative power in row classification, nearly doubling the F1 score of CM2 in the Movie domain (Table 2). These results confirm that consolidating lexical aliases into coherent schema representations produces more robust, linearly separable features than existing column-level baselines.
> > >
> > > (Table 2: Row Classification — F1 Score)
> > > ||Prod-XGB|Prod-CatB|Prod-LR|Mov-XGB|Mov-CatB|Mov-LR|
> > > |-|-|-|-|-|-|-|
> > > |CM2|0.814±0.023|0.805±0.013|0.846±0.018|0.379±0.025|0.370±0.015|0.375±0.020|
> > > |NAVI|**0.928±0.023**|**0.913±0.015**|**0.939±0.016**|**0.604±0.045**|**0.607±0.035**|**0.667±0.029**|
> > >
> > > ### Architectural Rationale
> > >
> > > Cell-level models (e.g., TaBERT, TABBIE, CM2) rely on intra-table contextualization($H_{\text{ctx}},V_{\text{ctx}}$), which inextricably links representations to specific row values and local structural noise. Aligning these purely contextualized cells forces the model to align table-specific artifacts, leading to "over-alignment". NAVI’s segment formulation resolves this through a specific architectural fusion:
> > > - `Distinction between Intent and Realization.` We treat a table *"header"* merely as a localized realization of a universal *"attribute"*. Unlike prior cell-centric models, NAVI's segment embedding explicitly incorporates a *context-free schema/attribute representation* ($H_{\text{ctx-free}}$) by encoding each of the attributes independently by using only itself as the context with a separate light-weight encoder. This embedding alone acts as a stable, universal reference point that is independent of any particular table row or distribution.
> > > - `The Alignment Mechanism.` In our Entropy-driven Segment Alignment (ESA), we do not simply align two "moving" contextualized representation. Instead, we align the variable instance-level realizations ($H_{\text{ctx}}, V_{\text{ctx}}$) and the context-free semantic anchors ($H_{\text{ctx-free}}$) utilizing the segment representation to mediate these representations. This process enriches the anchors with domain-specific semantics derived from local signals, allowing them to serve as a consistent basis for interpreting heterogeneous tables across the same domain.
> > >
> > > Crucially, this architecture fundamentally differs from prior cell-level approaches that aggregate only $H_{\text{ctx}}$ and $V_{\text{ctx}}$; by defining the segment as the integration of $H_{\text{ctx-free}}, H_{\text{ctx}}, V_{\text{ctx}}$, we provide the model with an explicit grounding signal that remains invariant to local table noise. The necessity of this grounding is also proven in Table 3 in our manuscript: while performance of models relying on table-specific cues degrade (-5.61% to -17.99%) under structural perturbations, NAVI remains robust (-1.28% to -6.20%). This demonstrates that $H_{\text{ctx-free}}$ is essential for successfully decoupling stable domain logic from entity-specific content.
> > >
> > > We hope our responses fully address your follow-up question. We thank you again for your feedback, which indeed helped us further highlight and clarify our contributions. We will fully incorporate these points in the final version.
> > >
> > > [1] Ye, Chao, et al. "Towards cross-table masked pretraining for web data mining." Proceedings of the ACM Web Conference 2024.

---

### Official Review · Reviewer_bzxy · 2026-03-13

**Soundness:** 2
**Presentation:** 2
**Significance:** 2
**Originality:** 3
**Overall Recommendation:** 4
**Confidence:** 4

**Summary:**

This paper proposes NAVI: Entropy-aware Alignment via Header–Value Induction. NAVI captures the structural properties of tables through schema-aware segment induction and modeling. In addition, NAVI employs entropy-driven alignment of segments to selectively incorporate domain knowledge shared among in-domain tables. Through various experiments, the paper shows effectiveness of NAVI on various downstream tasks.

**Compliance With Llm Reviewing Policy:**

Affirmed.

**Final Justification:**

The authors have addressed many of my concerns with many justifications. The enriched manuscript with discussions included, would provide interesting insights on table understanding but not fully adequate on predictive tabular learning.

**Key Questions For Authors:**

-	What is the ‘divergence in schema’ in figure 1? Also, how does the ‘shared domain semantics represented on the left? (Possibly, it would be nicer to have more specific wordings within the figure)
-	While this might not be of great interest for general tasks, how would NAVI address the problems of similar semantics, but totally different entities for low-cardinality? For example, city Paris can be the capital of France or it can be a city in Texas, and considering names to be of low-cardinality I think city can also be considered as of low-cardinality.
-	From the information provided from the paper, the baseline TableVectorizer uses the TextEncoder, which uses a language model under the hood. How would NAVI compare to TableVectorizer with StringEncoder, which uses simple tf-idf?
-	The TabPFN version, if possible, should be updated.
-	How important are the semantic information on the datasets? For instance, how would all the models perform with numerical values omitted?
-	Are there more datasets where performance of NAVI can be measured?
-	What is the hyperparameter spaces for the machine learning models? (Most likely, XGBoost, and CatBoost)

**Limitations:**

Yes.

**Strengths And Weaknesses:**

-	In general, the paper is easy to follow.
-	The wordings should be clarified for the readers. For example, ‘divergence in schema’ is hard to interpret.
-	The experiment results might be subjected for a specific set of benchmarks and might be difficult to grasp the general usability of the proposed method.
-	The results should also have standard deviation (or errors) to fully justify the superiority of NAVI.

---

> ### Author Rebuttal · Authors · 2026-03-31
>
> We thank reviewer bzxy for their careful reading and constructive feedback. We appreciate the recognition of the clarity of our framework and the novelty of the segment-based formulation. We address concerns on (i) terminology clarity, (ii) evaluation scope, and (iii) experimental details with additional clarification and analysis.
>
> ### SW2, Q1 - Clarification of Terminology
>
> By *'divergence in schemas and values'*, we refer to the lack of one-to-one alignment between schema elements (headers) and cell values across tables, even within the same domain. The same semantic role may appear under different headers (e.g., director vs. auteur), headers may be ambiguous (*name* referring to a movie or an actor), and values may correspond to different attributes depending on context (e.g., Paris as a city or a capital).
>
> To clarify the notion of *'shared domain semantics'*, we adopt a bottom-up view: attribute meaning is inferred from its empirical value distribution, while domain semantics emerge from consistent attribute co-occurrence patterns. In Fig. 1, although surface schemas differ, both tables exhibit similar value distributions and contextual attributes, indicating the same Movie domain. We will revise the manuscript to explicitly clarify this concept.
>
> ### Q2 - Entities with Similar Semantics
>
> We appreciate this important observation, as low-entropy attributes can indeed be ambiguous in real-world tables (e.g., the same value like “Paris” can correspond to different entities). Discriminative attributes are not fixed a priori, but determined by their empirical cell value distribution within a table. In our work, entropy is utilized as a soft inductive bias. If a column such as city name is low-entropy, it reflects relative stability, while disambiguation is handled by high-entropy attributes (e.g., country), distinguishing cases such as Paris, France and Paris, Texas. More generally, segment embeddings jointly consider context-free signals from attributes and contextualized signals from the row context, enabling disambiguation through interactions across attributes rather than a single column.
>
> Thus, our work does not collapse ambiguous values, addressing the concern that low-entropy attributes may lead to incorrect entity merging.
>
>
> ### SW3, Q5, Q6 - Evaluation Scope and Feature Analysis
>
> `Dataset Diversity.` This work targets representation learning under in-domain schema heterogeneity, reflecting realistic multi-table settings, not necessarily indicating universal cross-domain generalization. Thus, we focus on two representative text-heavy domains from Movie and Product domains, each containing **100 heterogeneous tables** from WDC WebTables, a large-scale real-world corpus with substantial schema variation even within a domain.
>
> `Text–Numeric Ratio Analysis.` We perform input ablation study with: (i) All Columns, (ii) Text-only, and (iii) Partial-text (half of text columns). Using XGBoost (8 subsampled runs), our work achieves 0.928±0.010, 0.902±0.014, and 0.888±0.015, respectively. Removing numerical columns leads to only modest changes, while reducing text coverage causes larger degradation. This indicates that semantic information in text columns is a primary driver of performance, consistent with modeling semantics jointly conditioned on schema context and value distributions.
>
> ### SW4, Q4, Q7 - Experimental Details
>
> **All results are averaged over multiple runs with different seeds**. While minor variance exists, the standard deviations remain narrow enough to confirm a consistent performance hierarchy, with our method maintaining a clear and statistically distinct lead over the baselines. For instance, representative statistics on the Product domain (using XGBoost) are: BERT 0.912±0.013, TAPAS 0.911±0.017, HAETAE 0.915±0.010, and NAVI(Ours) 0.928±0.010. We will update these in the manuscript.
>
> We use the recent TabPFN release at the time of submission, with the latest TabPFN-2.5 weights. For downstream classification, we adopt fixed configurations for fair comparison: XGBoost with eval_metric=mlogloss and tree_method=hist, and CatBoost with 100 iterations and default settings.
>
> ### Q3 - TableVectorizer with StringEncoder
>
> Following your suggestion, we evaluated TF-IDF-based StringEncoder and it achieves 0.919±0.017 (Product) and 0.524±0.014 (Movie). Our performance was comparable on Product (0.928, rising to 0.944 with simple numerical feature cocatenation (i.e., NAVI_FE)), and substantially outperforms on Movie (0.604/0.629(NAVI_FE)).
>
> This aligns with Sec. 5.4: TF-IDF-based representations suffice in cleaner schemas (Product), but struggle under schema redundancy and ambiguity (Movie, e.g., actor.1, actor.2). Moving beyond surface-level signals, TableVectorizer with TextEncoder benefits from generic semantic representations, while our work further models semantics conditioned on schema context and value distributions, enabling more robust performance across heterogeneous tables.

---

> > ### Author Rebuttal · Reviewer_bzxy · 2026-04-04
> >
> > Thanks to the authors for the response. The rebuttal has addressed many of my concerns.

---

> > > ### Author Response · Authors · 2026-04-07
> > >
> > > We sincerely thank the reviewer for the positive evaluation and for the helpful discussion. We are glad that our rebuttal fully addressed your concerns. As noted in your final justificaiton, we agree that this work opens a new perspective on understanding tabular data through a novel segment(header-value)-driven learning approach, thereby providing valuable insights to the relevant field of study and applications. Please kindly note that predictive tasks represent one of the key downstream applications of our work, alongside generative tasks. We believe the core conribution of our work lies in the effective representation of heterogeneous in-domain tables, enabling seamless integration with existing tabular methods, such as TableVectorizer, which has proven effective in further boosting predictive performance in conjunction with our work. Thank you again for your valuable time and your positive evaluation of our work. We will make sure to incorporate all discussed points into the final version of the paper.

---

### Decision · Program_Chairs · 2026-04-30

**Decision:**

Accept (regular)

**Comment:**

This paper introduces NAVI, a model for heterogeneous tabular representation that leverages segment embeddings and entropy-driven segment alignment to capture both schema-level consistency and instance-level variability. During the discussion phase, the reviewers reached a consensus for acceptance, with Reviewer xtRM specifically increasing their final recommendation to a "Weak accept". In their final justification, Reviewer xtRM noted that the authors effectively addressed their initial concerns , although they pointed out that NAVI has a larger model complexity (151M parameters) and its performance gains compared to baselines might be partly due to this increased capacity.
For the camera-ready version, the authors must meticulously incorporate all revisions promised during the rebuttal period. Specifically, they are required to explicitly define all mathematical terms and consistently distinguish between the concepts of "segment" and "segment embedding" in the text. Additionally, the authors must include an Impact Statement. Finally, they are expected to fully integrate the new empirical comparisons against more recent baselines—specifically TABBIE and CM2 (including the detailed value imputation and row classification results)—alongside the architectural clarifications discussed with the reviewers.
To sum up, the final AC decision is **accept** .